Registered report

psychology

sleep, memory, eyewitness identification, discriminability, confidence–accuracy relationship, line-ups

**Authors for correspondence:**
D. P. Morgan
e-mail: david.morgan@zi-mannheim.de
L. Mickes
e-mail: laura.mickes@bristol.ac.uk

†Present address: Central Institute of Mental Health, J5, 68159 Mannheim, Germany.

# The impact of sleep on eyewitness identifications

D. P. Morgan[1,2,3,†], J. Tamminen[4], T. M. Seale-Carlisle[5] and L. Mickes[6,7]

[1]Department of Clinical Psychology, [2]Department of Addiction Behavior and Addiction Medicine, and [3]Department of Psychiatry and Psychotherapy, Central Institute of Mental Health, Medical Faculty Mannheim, University of Heidelberg, Mannheim, Germany
[4]Department of Psychology, Royal Holloway, University of London, Egham, UK
[5]Department of Psychology, University of Birmingham, Birmingham, UK
[6]Department of Psychology, University of Bristol, Bristol, UK
[7]Department of Psychology, University of California, San Diego, CA, USA

DPM, 0000-0001-8213-451X; LM, 0000-0002-8090-9753

Sleep aids the consolidation of recently acquired memories. Evidence strongly indicates that sleep yields substantial improvements on recognition memory tasks relative to an equivalent period of wake. Despite the known benefits that sleep has on memory, researchers have not yet investigated the impact of sleep on eyewitness identifications. Eyewitnesses to crimes are often presented with a line-up (which is a type of recognition memory test) that contains the suspect (who is innocent or guilty) and fillers (who are known to be innocent). Sleep may enhance the ability to identify the guilty suspect and not identify the innocent suspect (i.e. discriminability). Sleep may also impact reliability (i.e. the likelihood that the identified suspect is guilty). In the current study, we manipulated the presence or the absence of sleep in a forensically relevant memory task. Participants witnessed a video of a mock crime, made an identification or rejected the line-up, and rated their confidence. Critically, some participants slept between witnessing the crime and making a line-up decision, while others remained awake. The prediction that participants in the sleep condition would have greater discriminability compared to participants in the wake condition was not supported. There were also no differences in reliability.

## 1. Introduction

Based on a large body of neuroscientific and behavioural evidence, conventional wisdom holds that sleep helps people remember better. While sleeping, newly learned information strengthens and stabilizes (i.e. consolidates) as a result of continued processing occurring in neural circuits, which are well established to be critical for memory [1–3]. This has been often shown in behavioural

experiments in which the results show that participants who sleep between learning information and being tested on that information consistently outperform participants who do not [4–11]. This finding applies widely in the tests of recognition memory for words, objects, faces and locations [12–17]. Theoretically, this should also apply to memory for crimes. However, the role that sleep may play on eyewitness identification has yet to be empirically investigated.

During the course of a criminal investigation, once the police find a suspect, a line-up may be administered to an eyewitness. A line-up is a type of recognition memory test that helps police assess the likelihood that the suspect is indeed the offender. Line-up procedures generally involve presenting the eyewitness with the police suspect among fillers. Fillers are known to be innocent, and physically resemble the suspect or match the description of the offender [18,19]. Of the three possible responses, an eyewitness can identify the suspect, identify a filler, or identify no one; whether the suspect was identified is typically of most applied interest. In this regard, if the suspect in the line-up is guilty, and the eyewitness chooses him or her, then that is a correct identification, and if the line-up is rejected, then that is a miss. If, on the other hand, the suspect in the line-up is innocent, and the eyewitness chooses him or her, then that is a false identification, and if the line-up is rejected, then that is a correct rejection. To put another way, two outcomes are errors (misses and false identifications), and two outcomes are accurate (correct identifications and correct rejections).

Considerable research has been conducted on a variety of variables that affect eyewitness memory performance [19–52]. How these variables affect eyewitness identifications are of interest to two different types of decision makers: policymakers (such as police and crime commissioners and police chiefs) and triers of truth (such as judges, magistrates and jurors). Because they have different decisions to make, they are consumers of different types of analyses, both of which entail suspect identifications [53]. One type of analysis assesses discriminability, which is the ability for eyewitnesses to distinguish innocent from guilty suspects. Procedures that reduce false identifications and increase correct identifications are procedures that have better discriminability than those that do not. Instituting procedures that increase discriminability is in the remit of policymakers. The other type of analysis assesses reliability, which is the probability that the identified suspect is the offender. In assessing the likelihood that a defendant is guilty, reliability, not discriminability, is informative to triers of truth. While the different assessments of both discriminability and reliability use suspect identifications, it is possible for the outcomes to diverge.

A variable low in discriminability may give rise to high reliability. For example, a study in which exposure duration was manipulated; participants saw the target for either 5 s or 90 s [20]. Discriminability was unsurprisingly lower for participants in the 5 s condition compared with that for participants in the 90 s condition, but identifications made with high confidence were equally high in accuracy regardless of condition.[1] In other words, these identifications were reliable whether the target was seen for 5 or 90 s. Other variables that have also shown this pattern of results where discriminability is lower for one condition than another, but reliability is equivalent (for identifications made with high confidence), include sequential versus simultaneous line-ups [53], short versus long retention intervals [24], weapon present versus weapon absent [21,22], offenders and witnesses are of different versus same races [23] and a verbal description versus no verbal description was provided [54].

Unlike the cases above in which discriminability and reliability vary, a decrease in discriminability may also lead to a decrease in reliability, for example, a study in which US line-ups were compared to UK line-ups. US line-ups yielded higher discriminability and higher reliability than UK line-ups [55]. This pattern also emerged when show-ups (i.e. just the police suspect, no fillers, are presented) were compared to simultaneous line-ups. In these studies, both discriminability and reliability were higher when memory was tested on a line-up compared with that tested on a show-up [24,53]. Similarly, unfair line-ups, compared to fair line-ups, were lower in discriminability and reliability [25,32,53]. Thus, a variable low in discriminability can be high or low in reliability, and (although we know of no research showing this is the case) it is theoretically possible that a variable high in discriminability can be low in reliability [53]. If sleep impacts discriminability, then scheduling identification procedures at optimal times would be in the purview of policymakers. If sleep impacts reliability, then knowing if sleep and the quality of sleep that occurred between the crime and the ID would be of interest to triers of truth.

How sleep affects witnesses' ability to identify an offender from a line-up is currently unknown. While some attention has been paid to the effects of sleep deprivation on eyewitness memory [56,57], as far as we

---

[1]Focus is typically placed on identifications made with high confidence because these are the identifications that matter in the court of law.

know, only one study correlated aspects of sleep with recognition memory for a mock crime [58]. In this study, participants (i) watched a mock crime video; (ii) completed questionnaires about sleepiness, sleep quality and sleep duration, and (iii) were immediately tested on a recognition memory test of central and peripheral details of the video. Sleepiness and poor sleep quality were negatively correlated with the accuracy of the peripheral details, but not the central details. This result provides some evidence that sleep affects some aspects of eyewitness memory; however, participants were not asked to identify the perpetrator. Furthermore, confidence ratings were not collected, and hits and false alarms were not reported, and so the impact of sleep on discriminability and reliability remains unknown. Given the well-known beneficial effects of sleep on memory and the lack of knowledge about the impact of sleep on eyewitness identifications, investigations of this type are highly overdue.

The goal of the current study is to measure the impact of sleep on discriminability and reliability on experimental eyewitnesses. To do so, we combined the AM-PM PM-AM sleep design with a forensically relevant design. The AM-PM PM-AM design is commonly used to compare the effects of sleep versus wake on recognition memory [7,15–17]. In this design, participants are assigned to a wake (AM-PM) or sleep (PM-AM) condition. In a forensically relevant design, which is similar to a standard list-learning recognition memory test, participants take part in a study (encoding) phase and test (retrieval) phase. A video of a mock crime is viewed in the former, and memory for the target in the video is tested on a line-up (either target-present or target-absent) in the latter. Unlike in a standard list-learning recognition memory test where there are multiple trials per participant, there is only one trial per participant in a forensically relevant design (to mimic the experience of a real eyewitness). In our integrated design, participants in the wake condition took part in the study phase in the morning (AM) and in the test phase in the evening (PM). Participants in the sleep condition took part in the study phase in the evening (PM) and in the test phase in the morning (AM). To rule out any influence of time-of-day effects on discriminability and reliability, which may otherwise explain our findings, we included two circadian control conditions where both the study phase and the test phase occurred either in the AM or in the PM to assess potential time-of-day confounds.

On the basis of the existing sleep literature, we predicted that discriminability will be greater for those in the sleep condition compared to those in the wake condition. The previous literature on sleep and declarative memory show that sleep benefits are not explained by the test phase occurring at different time periods of day in the sleep and wake conditions (e.g. [17,59,60]), and so we predicted no difference between the AM control and PM control conditions for discriminability. Due to the lack of literature of the effects of sleep and time-of-day effects on reliability, we did not have sufficient grounds to make predictions about this measure. However, because few variables have an appreciable effect on high confidence–accuracy [61], it seems reasonable to suppose that the same might be true of sleep. This investigation is a first step toward understanding how sleep may influence eyewitness identification performance.

# 2. Material and Methods

## 2.1. Participants

We used a sample size similar to other forensically relevant experiments in which discriminability and reliability were assessed (i.e. $n = 1000$ per condition) [25,55]. Sensitivity analysis revealed that by setting the parameters to standard values of $\alpha = 0.0125$ (corrected for each hypothesis test $\alpha = 0.05/4 = 0.0125$) and $1 - \beta = 0.95$, $n = 1000$ per experimental condition can detect an effect size of $d = 0.18$ in a two-tailed test. Participants ($N = 4000$) were randomly assigned to one of four conditions: wake ($n = 1000$), sleep ($n = 1000$), AM control ($n = 1000$) and PM control ($n = 1000$).[2] Participants were also randomly assigned to a target-present or target-absent line-up. When 4000 participants took part, data collection ceased. Royal Holloway, University of London Research Ethics Committee approved this study.

Participants were recruited online using the work source site Amazon Mechanical Turk [62]. To ensure good quality data, participation was limited to those who have hit approval rates of at least 85%. Approval rates are based on the quality of the responses on past tasks. Participant location was restricted to the US to ensure that data collection took place across only four time zones (Eastern Standard Time, Central Standard Time, Mountain Standard Time and Pacific Standard Time).

[2]Participants were randomly assigned to either the AM control and wake condition or the PM control and the sleep condition depending on the time of the day they chose to participate.

Individuals had to meet the following criteria: age between 18 and 40 years; not currently diagnosed with any sleep disorder (e.g. insomnia, sleep apnoea), psychiatric disorder (e.g. depression, post-traumatic stress disorder), or neurological disorder (e.g. mild cognitive impairment, Alzheimer's disease) that may affect memory [63–67]; not work as a shift worker; have not travelled across time zones within two weeks prior to participating; and not currently taking any prescribed medication that may affect sleep or memory.

## 2.2. Materials

### 2.2.1. Pre-screening questions

The pre-screening questions included yes–no questions about the inclusion criteria listed above.

### 2.2.2. Video

The study material was a 35 s video featuring a young adult white male (the target) stealing a laptop and mobile phone from an empty office. The target's face was clear throughout the video.

### 2.2.3. Line-ups

The line-ups were six-person simultaneous photo line-ups. Target-present line-ups contained a photo of the target and five fillers, and target-absent line-ups contained six fillers. The fillers matched the description of the target, which were provided by participants ($n = 19$) who watched the video and answered questions about the target's appearance. The averaged descriptors were entered into the Florida Department of Corrections database [68], and photos of 100 individuals who matched were extracted and grey scaled. The target and fillers were randomized to appear in any of the six line-up positions.

### 2.2.4. Distractor task

The distractor task was anagram puzzles [25,26]. Participants tried to solve 50 anagrams of US states for 5 min. This task was relevant to participants in the control conditions; however, all participants completed this task for the sake of consistency. Distractor tasks are standard in forensically relevant designs to prevent rehearsal between witnessing the crime and making an identification [25,26].

### 2.2.5. Sleep questionnaires

*Sleep-related questions.* Participants in the wake condition were instructed not to nap between the study and test phases, and were asked at the beginning of the test phase whether they did nap. If they answered yes, their data were excluded from the analysis and were replaced, as napping has been shown to aid memory consolidation [69]. Participants in all conditions were asked when they went to sleep and woke up to estimate how long they slept prior to participating in the study phase. Participants in the sleep condition were asked those questions again for their night's sleep between the study and test phase.

*Stanford sleepiness scale (SSS).* The SSS is used to measure an individual's current sleepiness using a seven-point scale (1 = feeling active, vital, alert or wide awake; 7 = no longer fighting sleep, sleep onset soon, having dream-like thoughts) [70]. High scores indicate high levels of sleepiness, and low scores indicate low levels of sleepiness. The data collected from the SSS were used as a manipulation check (see §4.5) to ensure the sleep and wake conditions differed only on this variable and not on tiredness at the time of participation.

*Reduced morningness eveningness questionnaire (MEQr).* The MEQr measures the time of day when an individual is more alert (chronotype) by answering five questions about their sleep habits (e.g. 'At what time of the day do you feel you become tired as a result of need for sleep?') [71]. Low scores indicate evening types and high scores indicate morning types. The data collected from this instrument were used for exploratory analyses only.

*Epworth sleepiness scale (ESS).* The ESS measures the general level of daytime sleepiness. Participants rate their chance of dozing while completing eight everyday tasks on four-point scale (0 = would never doze, 1 = slight chance of dozing, 2 = moderate chance of dozing and 3 = high chance of dozing) [72]. ESS scores range from 0 to 24. Low scores indicate normal levels of daytime sleepiness, and high scores

indicate high levels of daytime sleepiness. Participants who scored between 16 and 24 on the ESS (possibly indicating sleep difficulties) were excluded from the analysis and replaced.

*St Mary's Hospital sleep (SMHS) questionnaire*. The SMHS questionnaire measures duration and quality of the prior night's sleep [73]. Participants answered 'How well did you sleep last night?' by responding on a six-point scale (1 = very badly; 6 = very well). Low scores indicate poor sleep quality, and high scores indicate the high quality of sleep. The data collected using the SMHS questionnaire were used for exploratory analyses only.

## 2.3. Procedure

Participation took place online. Participants first answered the pre-screening questions. Those who did not meet the inclusionary criteria did not proceed with the experiment. In the study phase, participants digitally signed the consent form, answered the sleep-related questions (described in §2.2.5), completed the SSS, watched the mock crime video and completed the distractor task. In the test phase, participants tried to identify the target from the video on a line-up. Confidence ratings on a scale from 0 to 100% (0 = guessing, 100% = certain) were made for line-up decisions. Then, participants were asked if they have consumed any caffeine between the study and the test phase. This information was analysed as a part of our exploratory analyses. Next, participants completed the SSS, MEQr, ESS and the SMHS in a fixed order. Participants in the sleep condition answered the sleep-related questions two times, but participants in the wake and control conditions did not (as they did not sleep between the study and the test). After completing the sleep questionnaires and sleep-related questions, participants answered a validation question about the video (What was the crime committed in the video?) to assure that attention was paid during the study phase. If answered incorrectly, the data were excluded from the analysis.[3] The test phase, sleep questionnaires and validation questions were self-paced. Finally, participants were debriefed.

The order of the study and test phase was consistent across participants, but the timing varied. The procedure occurred at two different times for the participants in the sleep and wake conditions and only once for the control participants. Participants in the wake condition completed the study phase in the morning (between 08.00 and 11.00[4]) and completed the test phase approximately 12 h later in the evening (between 20.00 and 23.00). Participants in the sleep condition completed the study phase in the evening (between 20.00 and 23.00) and completed the test phase approximately 12 h later in the morning (between 08.00 and 11.00). Participants in the AM control condition completed the study and test phases in the morning (between 08.00 and 11.00), and participants in the PM control condition completed the study and test phases in the evening (between 20.00 and 23.00). The experiment was available for participation only during these hours.

# 3. Analysis strategy

Participants who incorrectly answered the validation questions, participants who reported napping between the study and test phases, participants who scored between 16 and 24 on the ESS (possibly indicating sleep difficulties) and participants who reported having less than 6 h of sleep [17] before the test phase were excluded from all analyses. Participants who were excluded were replaced to achieve the desired sample size (N = 4000). Alpha levels were set to 0.05, and Bonferroni corrections were used for multiple comparisons.

## 3.1. Correct and false ID rates

Correct ID rates were computed by dividing the number of guilty suspects identified by the total number of target-present line-ups. Because there is no designated innocent suspect, the false ID rate was estimated, which is standard practice [20]. The estimated false ID rates were computed by dividing the number of innocent suspects identified by the total number of target-absent line-ups divided by the number of line-up members (6).

---

[3]Previous research testing online with this type of design shows that less than 2% of participants answer this incorrectly (e.g. [57]).

[4]As approved by the editor during data collection after the stage 1 manuscript was accepted, the timings for data collection were extended from 08.00 and 10.00 and 20.00 to 22.00 to aid recruitment.

## 3.2. Discriminability

Receiver operating characteristic (ROC) analysis was conducted to measure discriminability [74,75]. The most common ROC approach is to collect eyewitnesses' confidence in their identifications, to plot correct ID rate and false ID rate pairs for every level of confidence and to measure the area under the ROC curve [76]. Because the false ID rate range extends from 0 to a value less than 1, we conducted the partial area under the curve (pAUC) analysis. To test our hypothesis that discriminability will be higher for those in the sleep condition compared to those in the wake condition, we compared the pAUC values from these conditions. These analyses were conducted using the pROC package in R [77].

To measure pAUCs, a false ID cut-off needs to be specified, and we used the most conservative overall false ID rate (i.e. we used the rightmost point on the ROC from the condition that yielded the more conservative responding overall), and so, the package did not have to extrapolate these points if a larger cut-off was used) [74]. The pROC package uses the number of suspect identifications for target-present line-ups and target-absent line-ups for every level of confidence and plots the empirical ROC as computed using trapezoids and are standardized with the formula

$$\frac{1}{2}\left(\frac{(1 + \text{pAUC} - \text{min})}{\text{max} - \text{min}}\right),$$

where min is the pAUC of chance responding and max is the pAUC of perfect performance. This was computed for each condition.

To compare the pAUC values, pROC derives $D$. $D$ is defined by

$$D = \frac{(\text{pAUC}_{\text{sleep}} - \text{pAUC}_{\text{wake}})}{s},$$

where $s$ is the standard error of the differences between the two pAUCs estimated by the bootstrap method (the number of bootstraps set to 10 000). The procedure or the variable with the greatest pAUC had the better discriminability.

## 3.3. Reliability

Confidence–accuracy characteristic (CAC) analysis was conducted to measure reliability [53]. For CAC analysis, for each level of confidence, the proportion correct was computed (e.g. high confidence guilty suspect IDs divided by the sum of high confidence correct IDs and high confidence estimated false IDs). A bootstrap procedure was used to estimate the standard errors associated with the suspect ID accuracy for each level of confidence for each condition. Observed data on target-present and target-absent line-ups were randomly sampled with replacement to obtain a bootstrap sample for each trial. This was repeated for 10 000 bootstrap trials, and the standard deviation of these trials yielded the estimated standard error. This was performed separately for each condition. Non-overlapping error bars signify difference between conditions [55].

## 3.4. Time-of-day effects

The control conditions were included to control for time-of-day effects. To determine whether time-of-day effects impact discriminability and reliability for eyewitness identifications, the ROCs and CACs for the AM control condition and the PM control condition were compared. To rule out differences in discriminability based on differences in circadian rhythms, we compared pAUC values of the two control conditions. Based on previous work, we did not expect these to differ [17,59,60]. ROC and CAC analyses were conducted using exactly the same procedure as outlined earlier. If no ROC and CAC differences existed between the AM and PM control conditions, then any differences in the experimental conditions would not be explained by time-of-day effects.

## 3.5. Current sleepiness

SSS values from the sleep and wake conditions at the study phase and the test phase were compared against each other as a manipulation check, using independent-samples $t$-tests (an $\alpha$-level set at 0.05, two-tailed test), to rule out the impact of sleepiness at encoding and retrieval. Based on the existing literature using the same AM-PM design, we did not expect statistically significant differences

**Table 1.** Demographic information by sleep, wake, AM control and PM control groups.

| | group | | | |
|---|---|---|---|---|
| | sleep | wake | AM control | PM control |
| gender | | | | |
| female | 586 | 604 | 614 | 612 |
| male | 410 | 391 | 386 | 384 |
| do not wish to state | 4 | 5 | 0 | 4 |
| age | | | | |
| years | 29.55 (5.71) | 30.40 (5.36) | 30.41 (5.31) | 29.43 (5.69) |
| ethnicity | | | | |
| African American | 70 | 67 | 85 | 94 |
| Arab American | 6 | 4 | 3 | 6 |
| Asian American | 85 | 63 | 59 | 67 |
| Caucasian American | 720 | 764 | 752 | 704 |
| Hispanic American | 65 | 48 | 56 | 61 |
| Indian American | 13 | 17 | 15 | 18 |
| Native American | 4 | 2 | 2 | 5 |
| do not wish to state | 11 | 11 | 10 | 15 |
| other | 26 | 24 | 18 | 30 |
| education | | | | |
| bachelor's degree | 425 | 450 | 415 | 461 |
| high school/GED | 67 | 83 | 87 | 66 |
| master's degree | 162 | 154 | 137 | 129 |
| post-master's | 60 | 60 | 50 | 53 |
| some college | 282 | 249 | 303 | 287 |
| some high school | 1 | 1 | 3 | 1 |
| do not wish to state | 3 | 3 | 5 | 3 |

between the conditions at either phase (e.g. [10,15–17]). Any remaining analyses were labelled as exploratory.

# 4. Results

The archived data are available in the electronic supplementary material, and the approved Stage 1 protocol is available at https://osf.io/x9j87. In total, data from 4309 participants were collected. Of those participants, 309 were excluded from all analyses for answering the validation question incorrectly ($n = 115$), napping between the study and the test ($n = 36$), having less than 6 h sleep ($n = 145$, $M = 7.59$ h, s.d. $= 0.96$, range $= 6$–12 h) or scoring more than 15 on the ESS ($n = 27$).[5] The data from the remaining 4000 participants were included in the analyses. Table 1 shows the demographic information.[6]

[5]This number does not sum to 309 because some participants could have been excluded for multiple reasons.

[6]A reviewer of the stage 2 manuscript recommended that we conduct analysis on differences in demographics between conditions. There were no significant differences in ethnicity ($\chi^2(12) = 1.55$, $p = 0.100$) ($\chi^2$ limited to ethnicities with cells $> 5$), gender ($\chi^2(3) = 0.61$, $p = 0.611$) or education ($\chi^2(12) = 1.61$, $p = 0.080$). There were significant differences in age between conditions ($F_{3,3996} = 9.18$, $p < 0.001$), but that is because the sample size is so large. The effect size is trivially small (0.007). An age difference of 0.85 (30.40–29.55 average years of the wake and sleep groups, respectively) is unlikely to account for results.

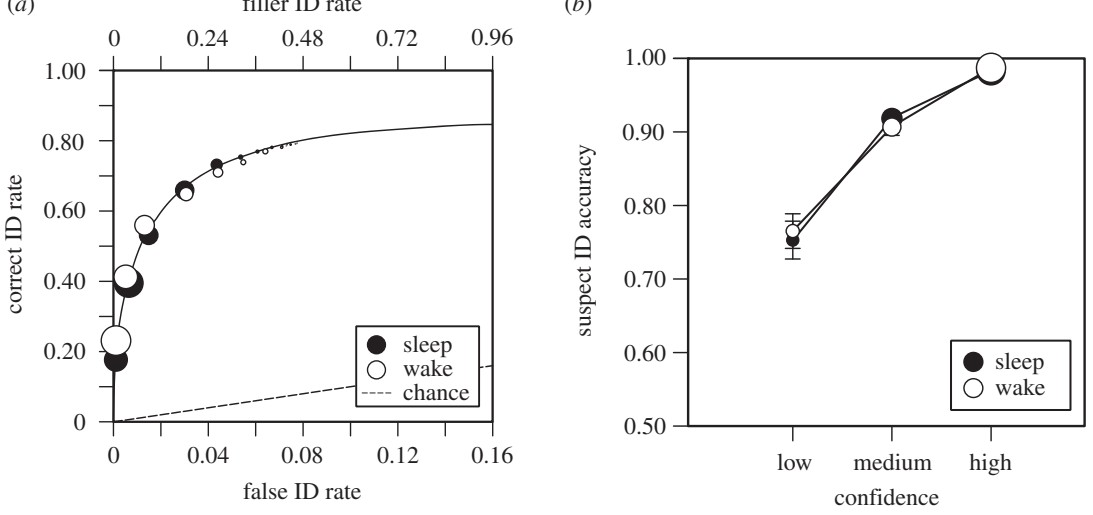

**Figure 1.** (*a*) ROC curves for sleep and wake groups. The operating points represent the data and the curves represent signal detection model fits. The overall filler ID rates from target-absent line-ups are shown on the top *x*-axis, and the estimated false ID rates are shown on the bottom *x*-axis. (*b*) CACs for sleep and wake groups. The size of the points represents relative frequencies of responses, and the bars represent standard errors.

**Table 2.** Frequencies of correct IDs (CIDs), filler IDs (FIDs) and no IDs for target-present and target-absent line-ups for each level of confidence for sleep and wake groups.

| | sleep | | | | | wake | | | | |
|---|---|---|---|---|---|---|---|---|---|---|
| | target-present | | | target-absent | | target-present | | | target-absent | |
| confidence | CID | FID | no ID | FID | no ID | CID | FID | no ID | FID | no ID |
| 0 | 2 | 0 | 3 | 1 | 4 | 1 | 2 | 3 | 2 | 11 |
| 10 | 0 | 4 | 1 | 5 | 2 | 1 | 1 | 0 | 6 | 2 |
| 20 | 2 | 3 | 3 | 13 | 4 | 4 | 1 | 3 | 11 | 9 |
| 30 | 6 | 7 | 7 | 18 | 15 | 6 | 10 | 6 | 20 | 10 |
| 40 | 8 | 5 | 9 | 21 | 15 | 16 | 4 | 3 | 27 | 8 |
| 50 | 11 | 2 | 10 | 30 | 27 | 15 | 5 | 9 | 31 | 16 |
| 60 | 37 | 11 | 9 | 40 | 37 | 32 | 11 | 12 | 39 | 37 |
| 70 | 65 | 6 | 14 | 45 | 70 | 46 | 3 | 10 | 51 | 52 |
| 80 | 69 | 2 | 3 | 25 | 51 | 76 | 5 | 8 | 23 | 51 |
| 90 | 111 | 2 | 2 | 16 | 28 | 94 | 0 | 5 | 12 | 36 |
| 100 | 90 | 1 | 2 | 3 | 23 | 120 | 1 | 4 | 3 | 26 |
| total | | 507 | | 493 | | | 517 | | 483 | |

## 4.1. Correct and false ID rates

Table 2 shows the frequencies of all of the response types by the level of confidence for target-present and target-absent line-ups for the sleep and wake conditions. The correct ID rates for both the sleep and wake conditions were 0.79. The estimated false ID rates for the sleep and wake conditions were 0.07 and 0.08, respectively.

## 4.2. Discriminability

Figure 1*a* shows the ROC curves for the sleep and wake conditions. The points represent the data and the size of the points reflects the relative frequencies of each [78]. All of the succeeding ROC and CAC

**Table 3.** Frequencies of correct IDs (CIDs), filler IDs (FIDs) and no IDs for target-present and target-absent line-ups for each level of confidence for AM and PM control groups.

| | AM control | | | | | PM control | | | | |
| | target-present | | | target-absent | | target-present | | | target-absent | |
| confidence | CID | FID | no ID | FID | no ID | CID | FID | no ID | FID | no ID |
|---|---|---|---|---|---|---|---|---|---|---|
| 0 | 0 | 0 | 2 | 4 | 2 | 1 | 1 | 4 | 6 | 2 |
| 10 | 0 | 2 | 0 | 9 | 5 | 1 | 7 | 1 | 2 | 5 |
| 20 | 3 | 5 | 0 | 14 | 4 | 3 | 1 | 1 | 12 | 5 |
| 30 | 7 | 2 | 1 | 13 | 17 | 7 | 5 | 0 | 20 | 11 |
| 40 | 9 | 2 | 3 | 28 | 18 | 8 | 1 | 2 | 20 | 16 |
| 50 | 14 | 6 | 10 | 31 | 22 | 13 | 2 | 9 | 37 | 27 |
| 60 | 32 | 1 | 5 | 27 | 39 | 24 | 5 | 1 | 34 | 33 |
| 70 | 46 | 5 | 11 | 29 | 61 | 70 | 4 | 8 | 30 | 54 |
| 80 | 73 | 1 | 5 | 26 | 57 | 74 | 6 | 4 | 14 | 56 |
| 90 | 91 | 0 | 1 | 5 | 59 | 95 | 0 | 3 | 8 | 49 |
| 100 | 146 | 0 | 3 | 1 | 43 | 140 | 1 | 2 | 6 | 49 |
| total | | 486 | | | 514 | | 504 | | | 496 |

figures feature relative frequencies. Figure 1a (and all ROC figures) are labelled with the overall filler ID rates from the target-absent line-ups (on the top x-axis) and estimated false ID rates (bottom x-axis) to show that either way to analyse the ROC data is legitimate [79]. The curves in figure 1a (and the ROC curves to follow) represent Ensemble unequal variance signal detection model fits [80]. By using a target-absent (TA) filler ID cut-off of 0.44, there was no difference between the pAUC values from the sleep (0.28) and wake (0.28) conditions, $D = 0.14$, $p = 0.892$. As shown in figure 1a, by the size of the points, the most common suspect identification for the wake condition was made with 100% confidence and the most common suspect identification for the sleep condition was made with 90% confidence.

## 4.3. Reliability

Because there are few responses in some of the confidence bins, as is clear in the rightmost points in figure 1a, we collapsed across confidence based on precedent [53] in the following manner: 0–60%, 70–80% and 90–100%. Figure 1b shows the CAC curves for the sleep and wake conditions. There were no differences in reliability, and in both cases, the majority of responses were made with high confidence, followed by medium confidence, followed by low confidence.

## 4.4. Time-of-day effects

Table 3 shows the frequencies of all of the response types by the level of confidence for target-present and target-absent line-ups for the AM and PM conditions. Correct ID rates for both the AM and PM conditions were 0.87, and the estimated false ID rates for both were 0.06. Figure 2a shows the ROC curves for the AM and PM conditions. By using a TA filler ID cut-off of 0.63, it was found that there was no difference between the pAUC values from the AM (0.278) and PM (0.271) conditions, $D = 0.69$, $p = 0.488$. Likewise, as shown in figure 2b, there were no differences between the AM and PM CAC curves. Thus, any differences that would have been found in the experimental conditions would not be explained by time-of-day effects. The AM and PM conditions yielded similar patterns of responses as the sleep and wake conditions, as reflected in the size of the points in figure 2a,b, whereby, in both conditions, suspects were identified most commonly with the highest level of confidence.

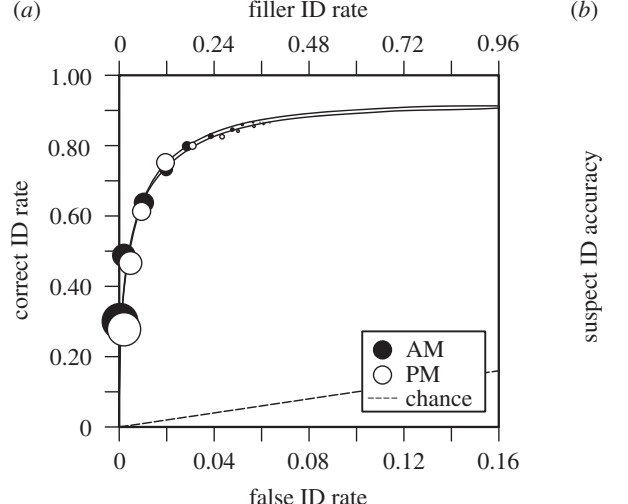 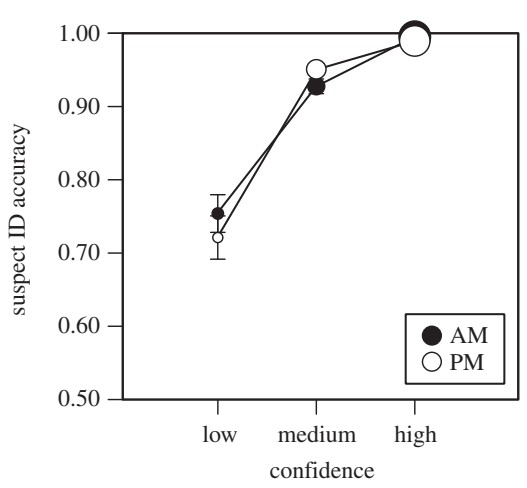

**Figure 2.** (*a*) ROC curves for AM and PM control groups. The operating points represent the data, and the curves represent signal detection model fits. The overall filler ID rates from target-absent line-ups are shown on the top *x*-axis, and the estimated false ID rates are shown on the bottom *x*-axis. (*b*) CACs for AM and PM control groups. The size of the points represents relative frequencies of responses, and the bars represent standard errors.

## 4.5. Current sleepiness

Across the sample, participants reported being sleepier in the evening compared with in the morning. The SSS scores were significantly higher for the sleep condition ($M = 2.70$, s.d. $= 1.19$) than the wake condition ($M = 2.09$, s.d. $= 1.01$) during the study phase (in the PM), $t_{1,998} = 12.27$, $p < 0.001$ and $d = 0.55$. The SSS scores were significantly higher for the wake condition ($M = 2.72$, s.d. $= 1.46$) than the sleep condition ($M = 2.28$, s.d. $= 1.18$) during the test phase (in the PM), $t_{1,998} = 7.31$, $p < 0.001$ and $d = 0.33$. The same pattern arises with the control conditions. Because participants in the control conditions report SSS scores at the same time point (unlike those in the experimental conditions), we averaged the two SSS scores from each participant in the control conditions, and those averages were significantly higher for the PM condition ($M = 2.52$, s.d. $= 1.19$) than the AM condition ($M = 2.10$, s.d. $= 0.99$), $t_{1,998} = 8.47$, $p < 0.001$ and $d = 0.38$.

## 4.6. Exploratory analyses

To rule out any impact that caffeine may have on performance [81], participants in the sleep and wake conditions who reported consuming caffeine were removed. We then conducted exploratory analyses on the remaining participants from the sleep ($n = 918$) and wake ($n = 887$) conditions. There was no discriminability difference between the pAUC of the sleep condition (0.28) versus the pAUC of the wake condition (0.29), $D = 0.53$ and $p = 0.593$ (a TA filler ID cut-off of 0.56 was used). There was also no reliability difference between the sleep and wake conditions. For the sleep condition, the proportions correct for identifications made with low, medium and high confidence were 75%, 92% and 98%, respectively. For the wake condition, the proportion correct identifications made with low, medium and high confidence were 76%, 91% and 99%, respectively. These results do not show support for changes in performance due to caffeine consumption.

To rule out any impact of poor sleep quality on performance, participants in the sleep condition who reported that they slept 'very badly' ($n = 3$), 'badly' ($n = 19$) or 'fairly badly' ($n = 113$) between study and test phases were removed. The remaining participants ($n = 865$) had a mean SMHS score of 5.14 (s.d. $= 0.52$, range $= 4$–$6$). We then compared the discriminability of this subset of participants in the sleep condition with those in the wake condition. There was no difference between the pAUC of the sleep condition (0.28) versus the wake condition (0.28), $D = 0.19$ and $p = 0.845$ (a TA filler ID cut-off of 0.56 was used). There was also no difference in reliability between the sleep and wake conditions. For the sleep condition, the proportion correct for identifications made with low, medium and high confidence was 74%, 92% and 98%, respectively. For the wake condition, the proportion correct for identifications made with low, medium and high confidence was 77%, 91% and 99%, respectively.

With the data from those who reported having poor sleep quality between the study and test phases removed, there were still no differences. The results in figure 1*a,b* cannot be explained by the poor sleep quality.

# 5. Discussion

We conducted a large-scale investigation on the impact that sleep may have on eyewitness identification performance, namely, discriminability and reliability. This is the largest sleep and episodic memory experiment to date. On the basis of the prior literature, we did not have strong predictions about how sleep would impact reliability, but we did predict that sleep would benefit the ability to discriminate guilty suspects from innocent suspects. The results did not bear out that prediction.

## 5.1. Discriminability

As predicted, there were no differences in discriminability between the time-of-day AM and PM control conditions. If there were differences between these control conditions, we would not be able to make strong claims about any differences in the experimental conditions. The data did not, however, support the prediction that discriminability was greater for the sleep condition versus the wake condition. Sleep did not improve memory. This result is counter to the many findings in the sleep literature (see reviews [1–3,82,83]). We consider and address several possible explanations for this discrepancy.

### 5.1.1. Differences in sleepiness levels

One possibility is that participants in the sleep condition were sleepier and therefore did not perform as well as they would have otherwise. We initially predicted no differences in sleepiness, based on the SSS scores, between the sleep and wake conditions or the AM and PM conditions [10,15–17], but there were differences. It was not the case, however, that the sleep condition uniformly reported being sleepier. Participants in all conditions reported being sleepier when asked in the evening. Participants in the sleep condition encoded the information in the evening when they reported being sleepier. If being sleepier at encoding affected discriminability, then the sleep condition may have had greater discriminability if they were not sleepy during the study phase.

Participants in the PM condition also reported being sleepier during the study phase, but there were no differences between the control conditions. Therefore, it is unlikely that the reason the participants in the sleep condition did not have greater discriminability than those in the wake condition is because they were sleepier during encoding. One could argue, however, that based on the encoding specificity principle [84], the PM controls, even though they were sleepy at encoding, did better because they were sleepy both at encoding and the test (i.e. same state of sleepiness), and for this reason, they performed with their AM counterparts. However, the state of sleepiness was not consistent for the participants in the experimental conditions.

To address this possible concern, we selected an equal number of participants from sleep and wake conditions who responded with a 1, 2 or 3 on the SSS on both occasions. By doing this, we screened out the participants who reported being sleepy (i.e. responded with a 4, 5, 6 or 7 on the SSS on both occasions) in the sleep ($n = 247$; $M = 1.87$ and s.d. $= 0.88$) and wake ($n = 247$; $M = 1.87$ and s.d. $= 0.88$) conditions and then compared discriminability. Although the sample sizes were reduced, we still had 95% power to detect a medium-sized effect, one that is comparable in the research literature on sleep (e.g. [6]). The pAUC for the sleep condition (0.23) was not significantly greater than the pAUC for the wake condition (0.20), $D = 1.32$ and $p = 0.188$ (with the TA filler ID pAUC cut-off of 0.62). Therefore, two lines of evidence suggest that the higher sleepiness does not explain the absence of a sleep benefit. First, participants in the PM condition were sleepier during the study phase, and this did not negatively impact their discriminability when compared with the participants in the AM control condition. Second, matching the levels of sleepiness between sleep and wake conditions during the study and test phases and comparing discriminability yielded no overall benefit of sleep on memory.

### 5.1.2. Differences in types of tests

Another possible reason that the sleep condition did not outperform the wake condition in the current experiment may be because the sleep benefit is more pronounced when memory is tested on a recall

test and less pronounced when memory is tested on a recognition test [85]. However, in a recent meta-analysis of over 40 published experiments, the effect sizes were medium when memory was tested on cued recall, free recall and recognition memory tests (0.55, 0.49 and 0.44, respectively) [86]. Therefore, we should be able to detect a sleep benefit even when using a recognition memory test. After all, our experiment has the power to detect a small effect.

Maybe it is just not the right type of recognition memory test. There are similarities and differences between standard list-learning and forensically relevant experiments. One similarity is that they both have target-absent and target-present trials. However, in list-learning experiments, targets and lures are shown one at a time and a decision is made on each item. When memory is tested on a line-up, the items are presented simultaneously, and one decision is made. Another difference is the number of trials. There are multiple trials per participant in list-learning experiments and only one trial per participant in forensically relevant experiments. Therefore, these types of recognition memory experiments differ in the relative influences of the sources of within- and between-subjects variance [80,87]. If the memory benefits afforded by sleep are conceptually replicable, then these differences should not account for the fact that in some of the list-learning experiments sleep enhanced memory, whereas there was no benefit of sleep in our experiment. Thus, differences in how memory is probed, in a free recall, cued recall or a type of recognition memory test, is an insufficient explanation for the lack of a sleep benefit in our experiment.

### 5.1.3. Comparing one forensically relevant experiment with another

Although we did not find an advantage of sleep on discriminability in a forensically relevant experiment, a recent paper reported one. In §1, we wrote that researchers had not yet investigated the impact of sleep on eyewitness identifications. However, this changed while our data collection was under way (for this registered report). A paper on the topic was published in which the findings of two forensically relevant experiments using an AM-PM design were reported [88]. In both experiments, participants watched a video of a mock crime, remained awake during the day or slept overnight, and then memory for the perpetrator was tested on a line-up. In one experiment, participants were only tested on target-present line-ups, and in the other experiment, participants were only tested on target-absent line-ups. In the former, there were no effects of sleep versus wake on correct identifications, filler identifications or misses. In the latter, there were effects of sleep versus wake on false identifications but not on correct rejections. There were fewer false identifications in the sleep condition.

To measure discriminability, both target-present and target-absent line-ups need to be included in the same experiment (e.g. [89–91]). Measuring correct ID rates and false ID rates separately has the potential to mislead (e.g. [92]). Lower false ID rates can indicate more conservative responding or increased discriminability. In the former, the lower false ID rate would be accompanied by a lower correct ID rate. This would not mean that memory is better. Instead it would mean that one condition is less likely to make an identification, so fewer innocent suspects would be identified, but fewer guilty suspects would be identified as well. That is the reason why participants should be tested on both target-present and target-absent line-ups in one experiment and why correct IDs and false IDs need to be considered together (e.g. [89,90]).

To assess whether we replicated the findings reported in the study by Stepan et al. [88], we conducted the same analyses on the responses from the target-present line-ups (a 2 (sleep, wake) × 3 (correct IDs, filler IDs, misses) $\chi^2$ analysis) and then on the responses from target-absent line-ups (a 2 (sleep, wake) × 2 (filler IDs, correct rejections) $\chi^2$ analysis). To match their sample sizes, we took the first 88 participants assigned to target-present line-ups in the wake ($n = 47$) and sleep ($n = 41$) conditions and the first 96 participants assigned to target-absent line-ups in the wake ($n = 53$) and sleep ($n = 43$) conditions. There were no differences between conditions in response types from the target-present line-ups, $\chi^2(2, N = 88) = 1.83$ and $p = 0.759$. There were also no differences between conditions in response types from the target-absent line-ups, $\chi^2(1, N = 96) = 0.014$ and $p = 0.904$. Thus, the results from the target-present line-ups replicated those reported in the study by Stepan et al., but the results from the target-absent line-ups did not.

When there is only one trial per participant, the sample size needs to be sufficiently large to accommodate for the fact that each participant provides only one data point. Because there were fewer than five responses in some of the cells, and $\chi^2$ analysis is sensitive to small sample sizes, we conducted the same analysis on our entire sample [93,94]. Again, there were no differences between conditions in response types from target-present line-ups, $\chi^2(2, N = 2000) = 0.321$ and $p = .571$, or target-absent line-ups, $\chi^2(1, N = 2000) = 0.65$, $p = 0.420$. These results map onto the results from the

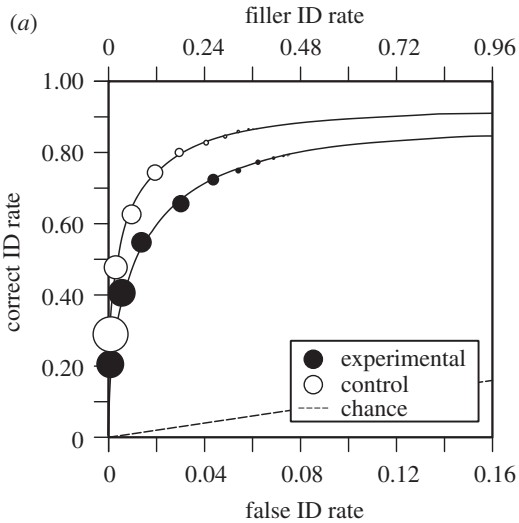
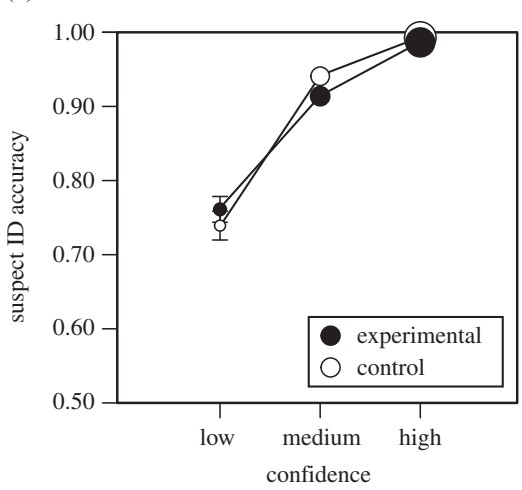

**Figure 3.** (*a*) ROC curves for experimental and control groups. The operating points represent the data, and the curves represent signal detection model fits. The overall filler ID rates from target-absent line-ups are shown on the top *x*-axis, and the estimated false ID rates are shown on the bottom *x*-axis. (*b*) CACs for experimental and control groups. The size of the points represents relative frequencies of responses, and the bars represent standard errors.

ROC analysis (a more suitable analysis than separate $\chi^2$ analyses) in which there were no differences between the experimental conditions.

Why did Stepan *et al.* [88] find a decreased false ID rate in the sleep condition compared to the wake condition and we did not? Our experiments were similar in that we used the same AM/PM design, memory was tested on a line-up and participants provided only one response. Despite these similarities, there were differences: we (i) did not use the same stimuli or procedures, (ii) we did not conduct separate experiments for target-absent and target-present line-ups, (iii) we conducted the experiment online and (iv) we had a sample size that was approximately 10 times larger. Therefore, maybe direct replication would find the same results. Another possible reason for the discrepancy between our experiments is that the lower false ID rate in the sleep condition in the study by Stepan *et al.* [88] was a false positive result [95].

## 5.2. Reliability

We did not make strong predictions about any effects that sleep may have on reliability (i.e. the likelihood that the suspect identified was indeed guilty). The sleep and wake conditions did not differ in reliability. Overall, the confidence that participants expressed was informative of accuracy. That is, averaged across all conditions, high confidence identifications were higher in accuracy than medium confidence identifications, which were higher in accuracy than low confidence identifications (99% versus 93% versus 75%, respectively).

One nuanced, and the important, concept is that even if discriminability is low, reliability can be high (e.g. [20,96]). Discriminability was lower for our experimental conditions compared to our control conditions. If this were not so, then we would have detected a problem because the experimental conditions were tested after a 12 h retention interval and the control conditions were tested after a 5 min retention interval. We conducted an exploratory analysis to determine whether reliability would be the same despite discriminability differing. Because there were no differences between the sleep and wake conditions and the AM and PM conditions, we collapsed across and compared the pAUC values of the experimental conditions with the control conditions. As shown in figure 3*a*, the pAUC for the control condition (0.276) was significantly greater than the pAUC for the experimental condition (0.229), $D = 5.92$ and $p < 0.001$ (with the false ID pAUC cut-off of 0.63).

Despite differences in discriminability between the experimental and control conditions, identifications made with high confidence were highly accurate (averaged across all conditions, they were 99% accurate). Figure 3*b* shows the CAC curves for the experimental and the control conditions. Thus, even though discriminability was lower for the experimental conditions than the control conditions (because there was a 12 h retention interval versus a 5 min retention interval, respectively),

participants were similarly well calibrated. That is, in the experimental and control conditions, identifications made with high confidence are higher in accuracy than identifications made with medium confidence, which are higher in accuracy than identifications made with low confidence (and the standard error bars are overlapping). This finding joins other findings in which even for lower discriminability conditions, reliability remains strong [61,96].

## 5.3. Conclusion

Sleep joins the growing list of variables that do not impact experimental eyewitnesses' reliability [53,61]. It also did not affect discriminability. Thus, the widely touted benefits of sleep were not found in a large-scale forensically relevant eyewitness identification experiment. Ours is not the first published experiment to find evidence that differs from most reports in the research literature of sleep [97,98]. Likewise, we failed to conceptually replicate the general findings reported in the literature that sleep benefits recognition memory. These findings challenge the widely claimed advantage that sleep has on recognition memory over a 12 h period. Further well-powered conceptual and direct replications should clarify the benefits of sleep on recognition memory.

Ethics. This study was approved by Royal Holloway, University of London Ethics committee (Project Number: Full-Review-295-2019-08-27-16-20). Informed consent was given by all of the participants.

Data accessibility. The archived data are available in the electronic supplementary material and the approved Stage 1 protocol is available at https://osf.io/x9j87.

Authors' contributions. D.P.M. created the stimuli and programmed the experiment. D.P.M. and T.M.S.-C. collected the data. D.P.M., T.M.S.-C. and L.M. conducted the analyses. All authors contributed to the conception and design of this experiment, wrote the manuscript and approved the final version of the manuscript for submission.

Competing interests. We declare we have no competing interests.

Funding. This work was supported in part by an Economic and Social Research Council (grant no. ES/L012642/1) to L.M. and by an Economic and Social Research Council (grant no. ES/P001874/1) to J.T.

Acknowledgements. We thank Kaloyan Ganev and Francis Lammas for acting in and helping to develop the video.

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
