## [Reviewer comments · Royal Society Open Science]

Review History

RSOS-170252.R0 (Original submission)

Review form: Reviewer 1

Is the language acceptable?

Yes

Do you have any ethical concerns with this paper?

No

Have you any concerns about statistical analyses in this paper?

No

Recommendation?

Accept in principle

Comments to the Author(s)

1. The significance of the research question(s)

This research seeks to answer an important, unanswered question. Although related research suggests that sleep should benefit discriminability, the question has not been asked about a forensically-relevant eyewitness experiment. And given the very few factors that appear to enhance eyewitness discriminability, potentially identifying another such variable like intervening sleep (or not), which could be easily accommodated by varying when an identification procedure is presented, is noteworthy. Also, as the authors point out, no one has examined the impact of sleep on the confidence-accuracy relationship (reliability). The vital role eyewitness confidence (on a first, fair test) can play in distinguishing accurate from inaccurate eyewitnesses is becoming increasingly clear in the literature

2. The logic, rationale, and plausibility of the proposed hypotheses

The experiment appears well-motivated and the hypotheses well-supported. The hypotheses are straightforward and ROC and CAC analyses will speak directly to these hypotheses.

3. The soundness and feasibility of the methodology and analysis pipeline (including statistical power analysis where applicable)

Mickes has vast experience with these types of forensically-relevant experiments, both methodologically, and analytically. The experiment is properly powered and the analyses are appropriate and properly address the hypotheses.

4. Whether the clarity and degree of methodological detail would be sufficient to replicate exactly the proposed experimental procedures and analysis pipeline

The methodology is described in sufficient detail and would allow a replication by interested parties. After publication, the data, and stimuli, should be shared on sites like <https://osf.io/OSF>.

5. Whether the authors provide a sufficiently clear and detailed description of the methods to prevent undisclosed flexibility in the experimental procedures or analysis pipeline

The authors make clear their Primary analyses (ROC and CAC), and separately spell out a number of additional (Secondary) analyses that are planned. Pre-registration is not intended to stifle data exploration, and it appears that the authors have made clear that analyses conducted beyond the Primary and Secondary will be properly declared Exploratory.

6. Whether the authors have considered sufficient outcome-neutral conditions (e.g. positive controls) for ensuring that the results obtained are able to test the stated hypotheses

Two time-of-day control conditions are included.

I like the study, I think it is important, and I am interested in what we will learn from it. I recommend that this research be accepted as a Registered Report. My only suggestion involves whether any of the sleep questionnaires collect information about caffeine consumption, and if not, whether that would be a worthwhile addition.

Review form: Reviewer 2

Is the language acceptable?

Yes

Do you have any ethical concerns with this paper?

No

Have you any concerns about statistical analyses in this paper?

No

Recommendation?

Accept with minor revision

Comments to the Author(s)

This is a very interesting study tackling a novel and important topic. Although the question whether eyewitness identification is affected by sleep is a very relevant practical issue, this question has not been formally tested yet. Findings on this question would also inform our general understanding of sleep and memory processes. The authors propose to examine two independent measures of eyewitness identification, i.e. discriminability and reliability. For discriminability they propose a specific hypothesis, i.e. they expect discriminability to be higher after sleep compared to a wake interval. This hypothesis is reasonable and based on previous evidence from sleep and recognition studies. However, for reliability the authors do not have a specific hypothesis based on lacking prior evidence. I would like to encourage the authors to state a hypothesis anyway, based on theoretical/conceptual considerations.

The proposed methods are adequate considering that this is a first broad study on the topic. Therefore it seems fine to obtain only very basic self-report information about sleep quality. However, if the authors want to draw sleep-specific conclusions, it would be important to test this question in a well-controlled environment with more fine-grained sleep measures, i.e. in a sleep lab with polysomnography. However, in a first step, as suggested in the present study, this will not be necessary, but the authors should be aware of the limited informational value of the sleep data they can obtain with questionnaires and this issue should also be discussed in the final paper.

The proposed statistical analyses are appropriate and well-established. Power estimations are difficult to run due to lacking previous evidence, but in my view the envisaged sample size seems sufficiently strong to detect at least medium size effects. The overall description of the methods is sufficiently clear for the authors not to use other undisclosed analyses or methods. However, I believe that minor changes that come up in the process of study preparation or data collection or analysis would be fine and can even be necessary.

The authors propose adequate controls for potential confounds and influencing factors. The two circadian (AM and PM) control groups are well suited to control for general circadian effects. Other relevant factors will be assessed via questionnaires, e.g. day-time sleepiness (ESS) and current sleepiness (SSS). This is all good. However, there are a few details that are not quite clear to me with regard to the control variables:

- Do the authors have a strategy for the case that there are circadian differences? How would the data in the sleep and wake groups be corrected?

- Moreover, I am not sure when exactly the current sleepiness (SSS) is going to be assessed. At some place in the procedures it sounds like the wake group would not fill out the SSS again during the test session. I would suggest obtaining two measures in the SSS in all groups (before study and after test).

- Related to the former point, the analysis strategy for the current sleepiness analysis should be refined. For this analysis, are the authors going to use the values from the study phase or test phase?

- Finally, does the SMHS sleep questionnaire have an overall score? If the authors intend to use the single sleep quality item only, this may not represent a valid measure of sleep quality. Other measures such as sleep duration, number of awakenings etc. should also be included in the sleep quality analysis.

The authors may consider introducing more exclusion criteria, e.g. psychiatric diagnoses (depression, PTSD?) and other medication (e.g. psychoactive drugs that may affect learning and memory). More questionnaires could be included to test for the exclusion criteria specifically instead of simply asking the subjects if these criteria apply to them. For example, the PSQI could be used to test for sleep disorders instead of simply asking "Are you currently diagnosed with a sleep disorder?" (many sleep disorders are not diagnosed and some subjects may not even know that they suffer from sleep disorders). Also, excluding subjects with less than 6 hours of sleep instead of 5 hours would be more reasonable.

Review form: Reviewer 3

Is the language acceptable?

Yes

Do you have any ethical concerns with this paper?

No

Have you any concerns about statistical analyses in this paper?

Yes

Recommendation?

Accept with minor revision

Comments to the Author(s)

SUMMARY

This proposed study is concerned with investigating the role of sleep in eyewitness identifications. Using measures of discriminability and reliability in a forensic identification setting, online participants will act as eyewitnesses through watching a video and attempt to identify the suspect in a subsequent photographic line-up. The interval between watching the video and line-up will contain either sleep or wake, with two further circadian control groups also present. A number of other sleep-related measures will be taken to investigate factors related to time of day, chronotype, sleep quality and sleepiness.

The proposed study is novel insofar as no eyewitness identification studies which look at performance as a function of sleep have been previously conducted, and it has clear real-world relevance. As such, it is a welcome study.

The design seems to be fundamentally sound, and most the proposed analyses are suitable for the intended purpose. The proposed sample size is large and the stimuli are appropriate. This

promises to be an interesting and informative study, and I see no fundamental reason why it should not go ahead.

I do, however, have a number of suggestions for potential improvements, and in particular requests for clarification of two key areas of the Introduction (including in relation to the key measures to be used in the study), addressing issues surrounding online participation, and giving careful thought to other aspects of the design including the False ID rate calculation and the randomisation of the questionnaires. These and other minor points are detailed below:-

SPECIFIC POINTS (page, line/section)

1. 1, 40: "Not identify the innocent suspect" – I think you mean innocent fillers; the fillers are not suspects.

2. 1, 56-57: "Sleep's role in shaping reliability is completely unknown and our work will be the first to explore this variable". As a broad statement (that does not mention identification specifically), that is not sustainable, even in the context of eyewitness testimony. See for, example Blagrove and Akehurst (2000) ("Effects of sleep loss on confidence-accuracy relationships for reasoning and eyewitness memory"), which is unquestionably about eyewitness reliability and the role of sleep, and which, incidentally, is a curious omission from the current paper which should be rectified, given its obvious relevance. It is important to give due respect to prior research and not overstate novelty.

3. 2, 51: These appears to be some confusion over terminology and possible outcomes. To be clear, there are two scenarios – a suspect is present and identified (hit) or a suspect is present and not identified (miss). The authors seem to believe the innocence or guilt of the suspect creates two further scenarios where an "innocent suspect" is present in the line-up. That raises two questions: (1) How would the eyewitness know the suspect is innocent or guilty? (2) What is the difference between an "innocent suspect" and a filler? Objectively, either the suspect was present and witnessed at the crime scene, or he wasn't. If he was, we're into the original two scenarios. If he wasn't, then how he is different to all the other fillers who were also not present at the crime scene? If he was present at the crime scene but did not commit the crime, how is that a different eyewitness scenario in terms of memory, to him being present at the crime scene and committing the crime? In both cases the identification requirement is the same. I wonder if in fact the intention for the two further scenarios is just a line-up full of fillers? If this is the case, then say it's a line-up full of fillers and don't suggest there is an "innocent suspect" amongst them, which is paradoxical. In that case, also explain the forensic relevance of this, given that the police do not routinely run identity parades with no known suspects present. In short, I don't think these four outcomes have been properly thought through, or at least are not properly explained here.

4. 3, throughout: The discussion about discriminability and reliability is similarly confusing and unconvincing. Discriminability is ability the "distinguish innocent from guilty suspects". Operationally, that means measuring the proportion of correct (vs incorrect) identifications, which is of course equivalent to the probability that the identified suspect is the offender. Reliability, on the other, we are told is "the probability that the identified suspect is the offender". In the other words, from the descriptions give, these are exactly the same measure. The examples given to illustrate that they are not the same measure don't help because they both use the same sleight-of-hand, comparing discriminability on all cases with reliability on only high-confidence cases and declaring that they are different. Of course they are different – you have used a different dataset for each. If you used the same data, they wouldn't be different, hence you have not demonstrated that these are different measures. If the measures are distinguished by the fact that one is only ever used in high-confidence cases while the other is always used in all cases, then the definitions should be given in those terms. As it is, I genuinely have no idea what, if

anything, is actually the difference from the definitions given in the text. They seem to be the same to me. Given that they are referred to separately throughout the study, this should certainly be rewritten to address the above concerns and clarify what the actual difference is.

5. 6, participants: A good number of participants are specified, and I understand the lack of existing effect sizes to guide this, but you can actually turn this around and indicate the effect size that can be achieved with given alpha and beta levels, e.g. at the usual 0.05 alpha rate and 95% power you could detect an effect size of $d=0.16$ in a direct comparison of sleep vs wake groups.

6. 6, 34-48: Online participation is a sensible choice for such a large sample size, but it brings with it a host of issues around reliability (and mturk has recently . You should outline what quality control measures are going to be used to address these. For example, you specify 18-40 inclusion; how will you know that people really are 18-40? What have previous online studies found in regard to reliability of age reports? Or, for example, you want people in AM and PM conditions; what checks will be undertaken on digital time stamps to ensure that they really were in those conditions? Similarly, what about issues in regard to the different demographics of participants on mturk compared to the wider population (highlighted in a number of studies, e.g. Casler et al, 2013)? These are all things to consider and address at the design stage; they shouldn't be just disregarded.

7. 6, 60: I would suggest excluding shift workers as well, who will have a sleep pattern notably different to other participants, and possibly individuals with neurological disorders which could affect memory performance.

8. 7, 53: The distractor task had not been mentioned until this section, so either mention it in the Introduction, or explain its purpose here.

9. 9, 18: It's entirely up to the authors, but I would suggest thinking about whether you really want to randomise the order the questionnaires. Insofar as there will be order effects, you are doing comparisons within dependent measures across groups, i.e. you have not proposed multivariate analyses where randomisation might be necessary. Instead, randomisation here will have the effect only of increasing noise. For example, if sleepiness is increased during the session, putting the SSS in different locations during the test session will increase the variability of those responses, but this variability will not reflect actual differences in sleepiness at the point of the identification test (which is the implied use of the SSS), but rather only the different location at which the SSS was measured. If there is no difference across the session, then randomisation would make no difference. Either way, there is no clear benefit, and a potential downside, so if it were me, I would keep the procedure, including test order, consistent across all participants.

10. 10, 27: If the estimated False ID rate procedure is standard, or at least had good precedent, then I would cite previous papers that have used it here. I would also point out that the measure has some undesirable properties. In particular, by dividing by the number of lineup members, it effectively weights a correct identification at six times an incorrect identification, which makes combining the Correct ID and False IS measures problematic. I understand the reason for doing this - it makes chance level the same for both measures - but the more you deviate from chance, the more unbalanced the measures become (e.g. 50% misses and 50% false alarms don't give anything like equivalent values). This will be an issue when plotting the ROC pairs and conducting that analysis. In some sense, this goes to the question of behavioural equivalence - is a correct identification equivalent to an incorrect identification, or six incorrect identifications, and the answer to that is as much related to usage as to statistics. It's something to give careful thought to at least, even if in the end the existing measure is still preferred.

Review form: Reviewer 4

Is the language acceptable?

Yes

Do you have any ethical concerns with this paper?

No

Have you any concerns about statistical analyses in this paper?

No

Recommendation?

Accept in principle

Comments to the Author(s)

The research question is highly important and will inform advice re treatment of victims and witnesses in terms of whether they are allowed to sleep before questioning.

The hypotheses are very plausible in terms of the considerable literature on effects of sleep on memory, and the results will have implications for that literature given that forensic-themed materials have been used only rarely in sleep and memory research.

The methodology and analyses are appropriate and are described in sufficient detail so as to allow exact replication.

The control condition of morning learning and evening testing is appropriate and a standard method, the methods will be able to test the proposed hypotheses.

Decision letter (RSOS-170252.R0)

20-Apr-2017

Dear Mr Morgan,

The Editors assigned to your stage one Registered Report ("The Impact of Sleep on Eyewitness Identifications") have now received comments from reviewers. We would like you to revise your paper in accordance with the referee and editors suggestions which can be found below (not including confidential reports to the Editor). Please note this decision does not guarantee eventual acceptance.

Please submit a copy of your revised paper within three weeks (i.e. by the 12-May-2017). If deemed necessary by the Editors, your manuscript will be sent back to one or more of the original reviewers for assessment.

To revise your manuscript, log into <http://mc.manuscriptcentral.com/rsos> and enter your Author Centre, where you will find your manuscript title listed under "Manuscripts with Decisions." Under "Actions," click on "Create a Revision." Your manuscript number has been

appended to denote a revision. Revise your manuscript and upload a new version through your Author Centre.

Kind regards,
Alice Power
Editorial Coordinator
Royal Society Open Science

on behalf of Chris Chambers
Registered Reports Editor, Royal Society Open Science
openscience@royalsociety.org

Associate Editor Comments to Author:

Comments to the Author:

Four expert reviewers have now assessed the manuscript. The reviews are broadly encouraging but note a number of key areas where greater clarity is required concerning rationale and hypotheses, study design and analysis. More thorough reporting of statistical power and proposed analyses is also required. As noted by Reviewer 3 (signed by Simon Durrant), the effect size that can be detected at a criterion level of power (e.g. 90% or 95%) can still be reported even in the absence of prior literature, e.g. using the Sensitivity Analysis option in G*Power. Please report this or a comparable analysis in the paper, taking into account the number of pre-registered hypothesis tests in the formulation of the alpha level (e.g. for five hypotheses, alpha would be .05/5).

In some case, it was unclear precisely which analyses will be conducted. Please ensure that specific tests are comprehensively described that map directly on to the proposed hypotheses. For example, more detail should be provided about the pROC procedure for inferring differences between conditions (Sec 4.2). In describing this procedure, it is not sufficient to refer solely to a previous publication because Registered Reports must be sufficiently internally comprehensive to permit replication. In later sections, the exact tests used to compare groups must be outlined.

In addition to these considerations, please also make clear whether excluded participants will be replaced; if not, the minimum acceptable sample size after exclusions should be pre-specified (and committed to achieving) with consequences for statistical power taken into account. More generally, we recommend reviewing the stated exclusion criteria to ensure that they are comprehensive, both for data within participants (if applicable) and participants within the sample, as these are not readily amendable after provisional acceptance.

A central feature of Registered Reports is the clear separation between the outcomes of pre-registered and exploratory analyses. Section 4.4 proposed a series of additional analyses but it is

unclear whether the authors intend these to be reported in the pre-registered section of the results at Stage 2 or instead whether they are loosely anticipating a series of post hoc analyses to be reported in the exploratory section of the results. If they are intended to be pre-registered, then they should be referred to as specific hypotheses, with full details about the statistical tests undertaken in each case. If, however, they are intended to be exploratory, then they should be removed from the Stage 1 submission and can be fully described and reported at Stage 2.

Comments to Author:

Reviewer: 1

Comments to the Author(s)

1. The significance of the research question(s)

This research seeks to answer an important, unanswered question. Although related research suggests that sleep should benefit discriminability, the question has not been asked about a forensically-relevant eyewitness experiment. And given the very few factors that appear to enhance eyewitness discriminability, potentially identifying another such variable like intervening sleep (or not), which could be easily accommodated by varying when an identification procedure is presented, is noteworthy. Also, as the authors point out, no one has examined the impact of sleep on the confidence-accuracy relationship (reliability). The vital role eyewitness confidence (on a first, fair test) can play in distinguishing accurate from inaccurate eyewitnesses is becoming increasingly clear in the literature

2. The logic, rationale, and plausibility of the proposed hypotheses

The experiment appears well-motivated and the hypotheses well-supported. The hypotheses are straightforward and ROC and CAC analyses will speak directly to these hypotheses.

3. The soundness and feasibility of the methodology and analysis pipeline (including statistical power analysis where applicable)

Mickes has vast experience with these types of forensically-relevant experiments, both methodologically, and analytically. The experiment is properly powered and the analyses are appropriate and properly address the hypotheses.

4. Whether the clarity and degree of methodological detail would be sufficient to replicate exactly the proposed experimental procedures and analysis pipeline

The methodology is described in sufficient detail and would allow a replication by interested parties. After publication, the data, and stimuli, should be shared on sites like <https://osf.io/OSF>.

5. Whether the authors provide a sufficiently clear and detailed description of the methods to prevent undisclosed flexibility in the experimental procedures or analysis pipeline

The authors make clear their Primary analyses (ROC and CAC), and separately spell out a number of additional (Secondary) analyses that are planned. Pre-registration is not intended to stifle data exploration, and it appears that the authors have made clear that analyses conducted beyond the Primary and Secondary will be properly declared Exploratory.

6. Whether the authors have considered sufficient outcome-neutral conditions (e.g. positive controls) for ensuring that the results obtained are able to test the stated hypotheses

Two time-of-day control conditions are included.

I like the study, I think it is important, and I am interested in what we will learn from it. I recommend that this research be accepted as a Registered Report. My only suggestion involves

whether any of the sleep questionnaires collect information about caffeine consumption, and if not, whether that would be a worthwhile addition.

Reviewer: 2

Comments to the Author(s)

This is a very interesting study tackling a novel and important topic. Although the question whether eyewitness identification is affected by sleep is a very relevant practical issue, this question has not been formally tested yet. Findings on this question would also inform our general understanding of sleep and memory processes. The authors propose to examine two independent measures of eyewitness identification, i.e. discriminability and reliability. For discriminability they propose a specific hypothesis, i.e. they expect discriminability to be higher after sleep compared to a wake interval. This hypothesis is reasonable and based on previous evidence from sleep and recognition studies. However, for reliability the authors do not have a specific hypothesis based on lacking prior evidence. I would like to encourage the authors to state a hypothesis anyway, based on theoretical/conceptual considerations.

The proposed methods are adequate considering that this is a first broad study on the topic. Therefore it seems fine to obtain only very basic self-report information about sleep quality. However, if the authors want to draw sleep-specific conclusions, it would be important to test this question in a well-controlled environment with more fine-grained sleep measures, i.e. in a sleep lab with polysomnography. However, in a first step, as suggested in the present study, this will not be necessary, but the authors should be aware of the limited informational value of the sleep data they can obtain with questionnaires and this issue should also be discussed in the final paper.

The proposed statistical analyses are appropriate and well-established. Power estimations are difficult to run due to lacking previous evidence, but in my view the envisaged sample size seems sufficiently strong to detect at least medium size effects. The overall description of the methods is sufficiently clear for the authors not to use other undisclosed analyses or methods. However, I believe that minor changes that come up in the process of study preparation or data collection or analysis would be fine and can even be necessary.

The authors propose adequate controls for potential confounds and influencing factors. The two circadian (AM and PM) control groups are well suited to control for general circadian effects. Other relevant factors will be assessed via questionnaires, e.g. day-time sleepiness (ESS) and current sleepiness (SSS). This is all good. However, there are a few details that are not quite clear to me with regard to the control variables:

- Do the authors have a strategy for the case that there are circadian differences? How would the data in the sleep and wake groups be corrected?
- Moreover, I am not sure when exactly the current sleepiness (SSS) is going to be assessed. At some place in the procedures it sounds like the wake group would not fill out the SSS again during the test session. I would suggest obtaining two measures in the SSS in all groups (before study and after test).
- Related to the former point, the analysis strategy for the current sleepiness analysis should be refined. For this analysis, are the authors going to use the values from the study phase or test phase?
- Finally, does the SMHS sleep questionnaire have an overall score? If the authors intend to use the single sleep quality item only, this may not represent a valid measure of sleep quality. Other measures such as sleep duration, number of awakenings etc. should also be included in the sleep quality analysis.

The authors may consider introducing more exclusion criteria, e.g. psychiatric diagnoses (depression, PTSD?) and other medication (e.g. psychoactive drugs that may affect learning and memory). More questionnaires could be included to test for the exclusion criteria specifically instead of simply asking the subjects if these criteria apply to them. For example, the PSQI could be used to test for sleep disorders instead of simply asking "Are you currently diagnosed with a sleep disorder?" (many sleep disorders are not diagnosed and some subjects may not even know that they suffer from sleep disorders). Also, excluding subjects with less than 6 hours of sleep instead of 5 hours would be more reasonable.

Reviewer: 3

Comments to the Author(s)

SUMMARY

This proposed study is concerned with investigating the role of sleep in eyewitness identifications. Using measures of discriminability and reliability in a forensic identification setting, online participants will act as eyewitnesses through watching a video and attempt to identify the suspect in a subsequent photographic line-up. The interval between watching the video and line-up will contain either sleep or wake, with two further circadian control groups also present. A number of other sleep-related measures will be taken to investigate factors related to time of day, chronotype, sleep quality and sleepiness.

The proposed study is novel insofar as no eyewitness identification studies which look at performance as a function of sleep have been previously conducted, and it has clear real-world relevance. As such, it is a welcome study.

The design seems to be fundamentally sound, and most the proposed analyses are suitable for the intended purpose. The proposed sample size is large and the stimuli are appropriate. This promises to be an interesting and informative study, and I see no fundamental reason why it should not go ahead.

I do, however, have a number of suggestions for potential improvements, and in particular requests for clarification of two key areas of the Introduction (including in relation to the key measures to be used in the study), addressing issues surrounding online participation, and giving careful thought to other aspects of the design including the False ID rate calculation and the randomisation of the questionnaires. These and other minor points are detailed below:-

SPECIFIC POINTS (page, line/section)

1. 1, 40: "Not identify the innocent suspect" - I think you mean innocent fillers; the fillers are not suspects.

2. 1, 56-57: "Sleep's role in shaping reliability is completely unknown and our work will be the first to explore this variable". As a broad statement (that does not mention identification specifically), that is not sustainable, even in the context of eyewitness testimony. See for, example Blagrove and Akehurst (2000) ("Effects of sleep loss on confidence-accuracy relationships for reasoning and eyewitness memory"), which is unquestionably about eyewitness reliability and the role of sleep, and which, incidentally, is a curious omission from the current paper which should be rectified, given its obvious relevance. It is important to give due respect to prior research and not overstate novelty.

3. 2, 51: These appears to be some confusion over terminology and possible outcomes. To be clear, there are two scenarios - a suspect is present and identified (hit) or a suspect is present and

not identified (miss). The authors seem to believe the innocence or guilt of the suspect creates two further scenarios where an “innocent suspect” is present in the line-up. That raises two questions: (1) How would the eyewitness know the suspect is innocent or guilty? (2) What is the difference between an “innocent suspect” and a filler? Objectively, either the suspect was present and witnessed at the crime scene, or he wasn't. If he was, we're into the original two scenarios. If he wasn't, then how he is different to all the other fillers who were also not present at the crime scene? If he was present at the crime scene but did not commit the crime, how is that a different eyewitness scenario in terms of memory, to him being present at the crime scene and committing the crime? In both cases the identification requirement is the same. I wonder if in fact the intention for the two further scenarios is just a line-up full of fillers? If this is the case, then say it's a line-up full of fillers and don't suggest there is an “innocent suspect” amongst them, which is paradoxical. In that case, also explain the forensic relevance of this, given that the police do not routinely run identity parades with no known suspects present. In short, I don't think these four outcomes have been properly thought through, or at least are not properly explained here.

4. 3, throughout: The discussion about discriminability and reliability is similarly confusing and unconvincing. Discriminability is ability the “distinguish innocent from guilty suspects”. Operationally, that means measuring the proportion of correct (vs incorrect) identifications, which is of course equivalent to the probability that the identified suspect is the offender. Reliability, on the other, we are told is “the probability that the identified suspect is the offender”. In the other words, from the descriptions give, these are exactly the same measure. The examples given to illustrate that they are not the same measure don't help because they both use the same sleight-of-hand, comparing discriminability on all cases with reliability on only high-confidence cases and declaring that they are different. Of course they are different – you have used a different dataset for each. If you used the same data, they wouldn't be different, hence you have not demonstrated that these are different measures. If the measures are distinguished by the fact that one is only ever used in high-confidence cases while the other is always used in all cases, then the definitions should be given in those terms. As it is, I genuinely have no idea what, if anything, is actually the difference from the definitions given in the text. They seem to be the same to me. Given that they are referred to separately throughout the study, this should certainly be rewritten to address the above concerns and clarify what the actual difference is.

5. 6, participants: A good number of participants are specified, and I understand the lack of existing effect sizes to guide this, but you can actually turn this around and indicate the effect size that can be achieved with given alpha and beta levels, e.g. at the usual 0.05 alpha rate and 95% power you could detect an effect size of $d=0.16$ in a direct comparison of sleep vs wake groups.

6. 6, 34-48: Online participation is a sensible choice for such a large sample size, but it brings with it a host of issues around reliability (and mturk has recently . You should outline what quality control measures are going to be used to address these. For example, you specify 18-40 inclusion; how will you know that people really are 18-40? What have previous online studies found in regard to reliability of age reports? Or, for example, you want people in AM and PM conditions; what checks will be undertaken on digital time stamps to ensure that they really were in those conditions? Similarly, what about issues in regard to the different demographics of participants on mturk compared to the wider population (highlighted in a number of studies, e.g. Casler et al, 2013)? These are all things to consider and address at the design stage; they shouldn't be just disregarded.

7. 6, 60: I would suggest excluding shift workers as well, who will have a sleep pattern notably different to other participants, and possibly individuals with neurological disorders which could affect memory performance.

8. 7, 53: The distractor task had not been mentioned until this section, so either mention it in the Introduction, or explain its purpose here.

9. 9, 18: It's entirely up to the authors, but I would suggest thinking about whether you really want to randomise the order the questionnaires. Insofar as there will be order effects, you are doing comparisons within dependent measures across groups, i.e. you have not proposed multivariate analyses where randomisation might be necessary. Instead, randomisation here will have the effect only of increasing noise. For example, if sleepiness is increased during the session, putting the SSS in different locations during the test session will increase the variability of those responses, but this variability will not reflect actual differences in sleepiness at the point of the identification test (which is the implied use of the SSS), but rather only the different location at which the SSS was measured. If there is no difference across the session, then randomisation would make no difference. Either way, there is no clear benefit, and a potential downside, so if it were me, I would keep the procedure, including test order, consistent across all participants.

10. 10, 27: If the estimated False ID rate procedure is standard, or at least had good precedent, then I would cite previous papers that have used it here. I would also point out that the measure has some undesirable properties. In particular, by dividing by the number of lineup members, it effectively weights a correct identification at six times an incorrect identification, which makes combining the Correct ID and False IS measures problematic. I understand the reason for doing this - it makes chance level the same for both measures - but the more you deviate from chance, the more unbalanced the measures become (e.g. 50% misses and 50% false alarms don't give anything like equivalent values). This will be an issue when plotting the ROC pairs and conducting that analysis. In some sense, this goes to the question of behavioural equivalence - is a correct identification equivalent to an incorrect identification, or six incorrect identifications, and the answer to that is as much related to usage as to statistics. It's something to give careful thought to at least, even if in the end the existing measure is still preferred.

Reviewer: 4

Comments to the Author(s)

The research question is highly important and will inform advice re treatment of victims and witnesses in terms of whether they are allowed to sleep before questioning.

The hypotheses are very plausible in terms of the considerable literature on effects of sleep on memory, and the results will have implications for that literature given that forensic-themed materials have been used only rarely in sleep and memory research.

The methodology and analyses are appropriate and are described in sufficient detail so as to allow exact replication.

The control condition of morning learning and evening testing is appropriate and a standard method, the methods will be able to test the proposed hypotheses.

Author's Response to Decision Letter for (RSOS-170252.R0)

See Appendix A.

Decision letter (RSOS-170501.R0)

18-May-2017

Dear Mr Morgan

On behalf of the Editor, I am pleased to inform you that your Manuscript RSOS-170501 entitled "The Impact of Sleep on Eyewitness Identifications" has been accepted in principle for publication in Royal Society Open Science.

You may now progress to Stage 2 and complete the study as approved. We would be grateful if you could now update the journal office as to the anticipated completion date of your study.

Following completion of your study, we invite you to resubmit your paper for peer review as a Stage 2 Registered Report. Please note that your manuscript can still be rejected for publication at Stage 2 if the Editors consider any of the following conditions to be met:

- The results were unable to test the authors' proposed hypotheses by failing to meet the approved outcome-neutral criteria
- The authors altered the Introduction, rationale, or hypotheses, as approved in the Stage 1 submission
- The authors failed to adhere closely to the registered experimental procedures
- Any post-hoc (unregistered) analyses were either unjustified, insufficiently caveated, or overly dominant in shaping the authors' conclusions
- The authors' conclusions were not justified given the data obtained

We encourage you to read the complete guidelines for authors concerning Stage 2 submissions at <http://rsos.royalsocietypublishing.org/content/registered-reports>. Please especially note the requirements for data sharing and that withdrawing your manuscript will result in publication of a Withdrawn Registration.

Once again, thank you for submitting your manuscript to Royal Society Open Science and I look forward to receiving your Stage 2 submission. If you have any questions at all, please do not hesitate to get in touch. We look forward to hearing from you shortly with the anticipated submission date for your stage two manuscript.

Kind regards,

Alice Power
Editorial Coordinator
Royal Society Open Science

on behalf of Chris Chambers
Registered Reports Editor, Royal Society Open Science
openscience@royalsociety.org

Author's Response to Decision Letter for (RSOS-170501.R0)

See Appendix B.

RSOS-170501.R1 (Revision)

Review form: Reviewer 1

Is the language acceptable?

Yes

Do you have any ethical concerns with this paper?

No

Have you any concerns about statistical analyses in this paper?

No

Recommendation?

Accept with minor revision

Comments to the Author(s)

The authors examined whether sleep can benefit eyewitness discriminability and reliability using a forensically-relevant experiment. Prior research on sleep (positive impact on memory for faces, locations) suggests it should benefit discriminability, but no prior work has examined reliability. If the authors had found positive impacts on discriminability, or differential effects on reliability, the findings would have been of practical significance. However, even the null effects here are informative, suggesting that the timing of a lineup test with regard to sleep does not appear to affect performance.

The authors made use of an AM-PM-PM-AM sleep design plus two circadian control conditions. The data were analyzed appropriately (and presented clearly) using ROC and CAC analyses. The conclusions are clear, if not terribly exciting. The fact that there was a significant difference detected in Figure 3 (control/immediate test vs. experimental/delayed test) shows that the authors had sufficient power to detect reasonably-size differences. The one concern I have is that overall performance is quite good. I am not concerned about ceiling effects, but if the authors had to do it over again (and I am NOT suggesting that), I would have preferred a little lower level of performance.

On p. 15 (and a couple of places thereafter), the authors referred to a false ID cutoff. I believe they meant filler ID cutoff.

On p. 23, the authors wrote: Despite these similarities, there were differences: we 1) did not use the same stimuli or procedures, 2) DID NOT conducted separate experiment (CAPS added). As written, makes it sound like the authors conducted separate experiments for target-present and target-absent, but it was Stepan et al. that did this.

1. Whether the data are able to test the authors' proposed hypotheses by passing the approved outcome-neutral criteria (such as absence of floor and ceiling effects or success of positive controls)

No floor or ceiling effects. Control groups appropriate and resulting data informative.

2. Whether the Introduction, rationale and stated hypotheses are the same as the approved Stage 1 submission

Yes

3. Whether the authors adhered precisely to the registered experimental procedures

Yes

4. Where applicable, whether any unregistered exploratory statistical analyses are justified, methodologically sound, and informative

I appreciated the exploratory statistical analyses that were conducted and found them all to be informative.

5. Whether the authors' conclusions are justified given the data

Yes

Review form: Reviewer 2

Is the language acceptable?

Yes

Do you have any ethical concerns with this paper?

No

Have you any concerns about statistical analyses in this paper?

No

Recommendation?

Accept with minor revision

Comments to the Author(s)

I have reviewed this manuscript before as a registered report prior to data collection. As already expressed in my first assessment of the manuscript, I feel that the study addresses a novel and important topic; the methods are sound and the proposed data analyses are appropriate. The authors have also included my previous suggestions on minor refinements of their protocol and analyses. This is all very good.

I would like to congratulate the authors on this comprehensive study, including such a large sample size. As far as I can see, the authors adequately adhered to their proposed experimental procedures as well as the planned statistical analyses. Beyond that, they have included appropriate exploratory analyses, which are clearly labelled as such.

The findings are very interesting in showing that, contrary to the hypothesis, sleep did not affect eyewitness identification (neither discriminability nor reliability). Given that the sample size was large enough to detect small effects, this is a very convincing demonstration that sleep may not influence the identification of perpetrators in lineups. These findings are very informative and practically relevant. The authors appropriately discuss possible explanations for this null finding and discuss potential implications.

There are only a few minor issues in the writing of the manuscript and the description of the methods that should be addressed in a revision:

1) I am not entirely clear on the style of a registered report paper. As is written now, the manuscript basically includes two separate parts, the first part including the introduction, methods and planned analyses as was submitted in the Stage 1 review. The second part includes the results and discussion sections, which were simply added to the first part. The two parts are clearly distinguishable, as the first part is written in future tense and the second in past tense. In the second part, some statements refer to other statements from the first part that have changed in the meantime (e.g. a new study on the same topic that has been published in the meantime). In my feeling, it would be more appropriate to re-write the entire paper as one single report, written in past tense. This would make the manuscript much more accessible for the reader. The abstract should also mention the results and a short conclusion statement.

2) The lineup procedure is not entirely clear to me. Was each subject presented with only one lineup? Did half of the subjects receive a target-present lineup and the other half a target-absent lineup? I was confused by the statement that "we... conducted separate experiments for target-

absent and target-present lineups" (page 23). Please clarify and describe more clearly in the methods section.

3) Please explain in more detail what the most conservative overall false ID rate in the ROC analyses refers to and how it was determined.

4) Table 1 should include statistical comparisons between the single groups to show that all groups were comparable in demographic features.

5) The terms "filler IDs" and "no-IDs" are mentioned in the results section and in the Tables and Figures, yet these terms are not introduced in the methods section. Are filler IDs identical to false IDs? Please clarify and include descriptions in the methods section.

6) With regard to the exploratory analyses, it would also be interesting to test for the effects of habitual sleepiness (ESS), chronotype and total sleep time in the experimental night. The authors may consider adding respective analyses.

7) In my view, the additional analyses in Figure 3 are particularly important since it is not a trivial question that discriminability declines across a 12 hour delay. This may be discussed in more detail as it is a practically relevant issue. It is also an interesting finding that reliability does not seem to deteriorate across time (at least across 12 hours). Please report statistics for the latter finding (page 23). The term "similarly well-calibrated" sounds unusual to me and could be rephrased.

8) Typos: "some aspects eyewitness memory" (page 5), "higher than accuracy then low confidence identifications" (page 23), in reference #63 two references seem to be mixed up, in Table 3 it should probably read "for AM and PM controls", in Figure 3B it should probably read "for experimental and control groups"

Decision letter (RSOS-170501.R1)

11-Oct-2019

Dear Mr Morgan:

On behalf of the Editor, I am pleased to inform you that your Stage 2 Registered Report RSOS-170501.R1 entitled "The Impact of Sleep on Eyewitness Identifications" has been deemed suitable for publication in Royal Society Open Science subject to minor revision in accordance with the referee suggestions. Please find the referees' comments at the end of this email.

The reviewers and Subject Editor have recommended publication, but also suggest some minor revisions to your manuscript. Therefore, I invite you to respond to the comments and revise your manuscript.

Please also ensure that all the below editorial sections are included where appropriate -- if any section is not applicable to your manuscript, please can we ask you to nevertheless include the heading, but explicitly state that the heading is inapplicable. An example of these sections is attached with this email.

- Ethics statement

- Data accessibility

[http://datadryad.org/submit?journalID=RSOS&manu=\(Document not available\)](http://datadryad.org/submit?journalID=RSOS&manu=(Document not available))

- Competing interests

- Authors' contributions

- Acknowledgements

- Funding statement

Because the schedule for publication is very tight, it is a condition of publication that you submit the revised version of your manuscript within 7 days (i.e. by the 19-Oct-2019). If you do not think you will be able to meet this date please let me know immediately.

Please note that Royal Society Open Science will introduce article processing charges for all new submissions received from 1 January 2018. Registered Reports submitted and accepted after this date will ONLY be subject to a charge if they subsequently progress to and are accepted as Stage 2 Registered Reports. If your manuscript is submitted and accepted for publication after 1 January 2018 (i.e. as a full Stage 2 Registered Report), you will be asked to pay the article processing charge, unless you request a waiver and this is approved by Royal Society Publishing. You can find out more about the charges at <http://rsos.royalsocietypublishing.org/page/charges>. Should you have any queries, please contact openscience@royalsociety.org.

Kind regards,
Anita Kristiansen
Royal Society Open Science
openscience@royalsociety.org

on behalf of Chris Chambers (Registered Reports Editor, Royal Society Open Science)
openscience@royalsociety.org

Associate Editor Comments to Author (Professor Chris Chambers):

Associate Editor: 1

Comments to the Author:

Two of the four expert reviewers who assessed the Stage 1 manuscript have now appraised the completed Stage 2 submission. The reviews are overall very positive, with only minor revisions recommended. Reviewer 1 requests some changes to resolve issues of clarity. Reviewer 2 recommends a shift from future to past tense in the Introduction and Method sections. This is an acceptable alteration and one that I would also recommend in the interests of readability (please ensure that all changes are tracked in the revised manuscript). The reviewer also suggests an additional analysis (Reviewer 2, point 6). Additional exploratory analyses are not required for Stage 2 acceptance, except where necessary to support the conclusions. In this case, these analyses do not seem necessary for the current conclusions to be appropriate, therefore it is up to the authors to decide if the analyses would be appropriate and informative. If so, the authors are welcome to conduct these extra analyses and report them in the Exploratory section of the Results.

Reviewer: 1

Comments to the Author(s)

The authors examined whether sleep can benefit eyewitness discriminability and reliability using a forensically-relevant experiment. Prior research on sleep (positive impact on memory for faces, locations) suggests it should benefit discriminability, but no prior work has examined reliability. If the authors had found positive impacts on discriminability, or differential effects on reliability, the findings would have been of practical significance. However, even the null effects here are informative, suggesting that the timing of a lineup test with regard to sleep does not appear to affect performance.

The authors made use of an AM-PM-PM-AM sleep design plus two circadian control conditions. The data were analyzed appropriately (and presented clearly) using ROC and CAC analyses. The conclusions are clear, if not terribly exciting. The fact that there was a significant difference detected in Figure 3 (control/immediate test vs. experimental/delayed test) shows that the authors had sufficient power to detect reasonably-size differences. The one concern I have is that overall performance is quite good. I am not concerned about ceiling effects, but if the authors had to do it over again (and I am NOT suggesting that), I would have preferred a little lower level of performance.

On p. 15 (and a couple of places thereafter), the authors referred to a false ID cutoff. I believe they meant filler ID cutoff.

On p. 23, the authors wrote: Despite these similarities, there were differences: we 1) did not use the same stimuli or procedures, 2) DID NOT conducted separate experiment (CAPS added). As written, makes it sound like the authors conducted separate experiments for target-present and target-absent, but it was Stepan et al. that did this.

1. Whether the data are able to test the authors' proposed hypotheses by passing the approved outcome-neutral criteria (such as absence of floor and ceiling effects or success of positive controls)

No floor or ceiling effects. Control groups appropriate and resulting data informative.

2. Whether the Introduction, rationale and stated hypotheses are the same as the approved Stage 1 submission

Yes

3. Whether the authors adhered precisely to the registered experimental procedures

Yes

4. Where applicable, whether any unregistered exploratory statistical analyses are justified, methodologically sound, and informative

I appreciated the exploratory statistical analyses that were conducted and found them all to be informative.

5. Whether the authors' conclusions are justified given the data

Yes

Comments to Author:

Reviewer: 2

Comments to the Author(s)

I have reviewed this manuscript before as a registered report prior to data collection. As already expressed in my first assessment of the manuscript, I feel that the study addresses a novel and important topic; the methods are sound and the proposed data analyses are appropriate. The authors have also included my previous suggestions on minor refinements of their protocol and analyses. This is all very good.

I would like to congratulate the authors on this comprehensive study, including such a large sample size. As far as I can see, the authors adequately adhered to their proposed experimental procedures as well as the planned statistical analyses. Beyond that, they have included appropriate exploratory analyses, which are clearly labelled as such.

The findings are very interesting in showing that, contrary to the hypothesis, sleep did not affect eyewitness identification (neither discriminability nor reliability). Given that the sample size was large enough to detect small effects, this is a very convincing demonstration that sleep may not influence the identification of perpetrators in lineups. These findings are very informative and practically relevant. The authors appropriately discuss possible explanations for this null finding and discuss potential implications.

There are only a few minor issues in the writing of the manuscript and the description of the methods that should be addressed in a revision:

1) I am not entirely clear on the style of a registered report paper. As is written now, the manuscript basically includes two separate parts, the first part including the introduction, methods and planned analyses as was submitted in the Stage 1 review. The second part includes the results and discussion sections, which were simply added to the first part. The two parts are clearly distinguishable, as the first part is written in future tense and the second in past tense. In the second part, some statements refer to other statements from the first part that have changed in the meantime (e.g. a new study on the same topic that has been published in the meantime). In my feeling, it would be more appropriate to re-write the entire paper as one single report, written in past tense. This would make the manuscript much more accessible for the reader. The abstract should also mention the results and a short conclusion statement.

2) The lineup procedure is not entirely clear to me. Was each subject presented with only one lineup? Did half of the subjects receive a target-present lineup and the other half a target-absent lineup? I was confused by the statement that "we... conducted separate experiments for target-absent and target-present lineups" (page 23). Please clarify and describe more clearly in the methods section.

3) Please explain in more detail what the most conservative overall false ID rate in the ROC analyses refers to and how it was determined.

4) Table 1 should include statistical comparisons between the single groups to show that all groups were comparable in demographic features.

5) The terms "filler IDs" and "no-IDs" are mentioned in the results section and in the Tables and Figures, yet these terms are not introduced in the methods section. Are filler IDs identical to false IDs? Please clarify and include descriptions in the methods section.

6) With regard to the exploratory analyses, it would also be interesting to test for the effects of habitual sleepiness (ESS), chronotype and total sleep time in the experimental night. The authors may consider adding respective analyses.

7) In my view, the additional analyses in Figure 3 are particularly important since it is not a trivial question that discriminability declines across a 12 hour delay. This may be discussed in more detail as it is a practically relevant issue. It is also an interesting finding that reliability does not seem to deteriorate across time (at least across 12 hours). Please report statistics for the latter finding (page 23). The term "similarly well-calibrated" sounds unusual to me and could be rephrased.

8) Typos: "some aspects eyewitness memory" (page 5), "higher than accuracy then low confidence identifications" (page 23), in reference #63 two references seem to be mixed up, in Table 3 it should probably read "for AM and PM controls", in Figure 3B it should probably read "for experimental and control groups"

Author's Response to Decision Letter for (RSOS-170501.R1)

See Appendix C.

Decision letter (RSOS-170501.R2)

14-Nov-2019

Dear Mr Morgan:

It is a pleasure to accept your Stage 2 Registered Report entitled "The Impact of Sleep on Eyewitness Identifications" in its current form for publication in Royal Society Open Science. The comments of the reviewer(s) who reviewed your manuscript are included at the foot of this letter.

on behalf of Professor Chris Chambers (Subject Editor)
openscience@royalsociety.org

Appendix A

Please see our responses to the editor and reviewers' comments printed in blue font below. In the manuscript text that has been changed or added is also printed in blue font.

Associate Editor Comments to Author:

Comments to the Author:

Four expert reviewers have now assessed the manuscript. The reviews are broadly encouraging but note a number of key areas where greater clarity is required concerning rationale and hypotheses, study design and analysis. More thorough reporting of statistical power and proposed analyses is also required. As noted by Reviewer 3 (signed by Simon Durrant), the effect size that can be detected at a criterion level of power (e.g. 90% or 95%) can still be reported even in the absence of prior literature, e.g. using the Sensitivity Analysis option in G*Power. Please report this or a comparable analysis in the paper, taking into account the number of pre-registered hypothesis tests in the formulation of the alpha level (e.g. for five hypotheses, alpha would be .05/5).

This is a great recommendation. By setting the parameters to standard values of $\alpha = 0.0125$ (corrected for each hypothesis test $\alpha = 0.05/4 = 0.0125$) and power of 95%, $n = 1000$ per experimental condition can detect an effect size of $d = 0.18$ in a two-tailed test. This analysis has been added to the revised manuscript (p. 7).

In some case, it was unclear precisely which analyses will be conducted. Please ensure that specific tests are comprehensively described that map directly on to the proposed hypotheses. For example, more detail should be provided about the pROC procedure for inferring differences between conditions (Sec 4.2). In describing this procedure, it is not sufficient to refer solely to a previous publication because Registered Reports must be sufficiently internally comprehensive to permit replication. In later sections, the exact tests used to compare groups must be outlined.

We now provide more information that links the hypotheses tests and their respective analyses (see Sec 4).

In addition to these considerations, please also make clear whether excluded participants will be replaced; if not, the minimum acceptable sample size after exclusions should be pre-specified (and committed to achieving) with consequences for statistical power taken into account. More generally, we recommend reviewing the stated exclusion criteria to ensure that they are comprehensive, both for data within participants (if applicable) and participants within the sample, as these are not readily amendable after provisional acceptance.

Excluded participants will be replaced to achieve the specified sample size, which has been added to the paper (p. 12). We also revised the pre-screening and exclusionary criteria following reviewers' advice to ensure they are comprehensive (p. 7-8 and p. 12).

In response to the concerns over the pre-screening and exclusionary criteria we have also limited the experiment to mTurk workers with a hit approval rating of at least 85%. This ensures that only good quality workers participate in this experiment (p. 7).

A central feature of Registered Reports is the clear separation between the outcomes of pre-registered and exploratory analyses. Section 4.4 proposed a series of additional analyses but it

is unclear whether the authors intend these to be reported in the pre-registered section of the results at Stage 2 or instead whether they are loosely anticipating a series of post hoc analyses to be reported in the exploratory section of the results. If they are intended to be pre-registered, then they should be referred to as specific hypotheses, with full details about the statistical tests undertaken in each case. If, however, they are intended to be exploratory, then they should be removed from the Stage 1 submission and can be fully described and reported at Stage 2.

We have removed the exploratory analyses (i.e. the impact of chronotype, general and current sleepiness and sleep quality on discriminability and reliability of identifications) that will be reported at Stage 2.

Reviewer: 1

Comments to the Author(s)

1. The significance of the research question(s)

This research seeks to answer an important, unanswered question. Although related research suggests that sleep should benefit discriminability, the question has not been asked about a forensically-relevant eyewitness experiment. And given the very few factors that appear to enhance eyewitness discriminability, potentially identifying another such variable like intervening sleep (or not), which could be easily accommodated by varying when an identification procedure is presented, is noteworthy. Also, as the authors point out, no one has examined the impact of sleep on the confidence-accuracy relationship (reliability). The vital role eyewitness confidence (on a first, fair test) can play in distinguishing accurate from inaccurate eyewitnesses is becoming increasingly clear in the literature

2. The logic, rationale, and plausibility of the proposed hypotheses

The experiment appears well-motivated and the hypotheses well-supported. The hypotheses are straightforward and ROC and CAC analyses will speak directly to these hypotheses.

3. The soundness and feasibility of the methodology and analysis pipeline (including statistical power analysis where applicable)

Mickes has vast experience with these types of forensically-relevant experiments, both methodologically, and analytically. The experiment is properly powered and the analyses are appropriate and properly address the hypotheses.

4. Whether the clarity and degree of methodological detail would be sufficient to replicate exactly the proposed experimental procedures and analysis pipeline

The methodology is described in sufficient detail and would allow a replication by interested parties. After publication, the data, and stimuli, should be shared on sites like <https://osf.io/OSF>.

Our data and stimuli will be made available on <http://datadryad.org> (the repository of Royal Society Open Science).

5. Whether the authors provide a sufficiently clear and detailed description of the methods to prevent undisclosed flexibility in the experimental procedures or analysis pipeline

The authors make clear their Primary analyses (ROC and CAC), and separately spell out a number of additional (Secondary) analyses that are planned. Pre-registration is not intended to stifle data exploration, and it appears that the authors have made clear that analyses conducted beyond the Primary and Secondary will be properly declared Exploratory.

6. Whether the authors have considered sufficient outcome-neutral conditions (e.g. positive controls) for ensuring that the results obtained are able to test the stated hypotheses
Two time-of-day control conditions are included.

The control conditions are included to assess the possibility that time of day differences may explain the results instead of sleep. In our original manuscript these details were briefly mentioned in a footnote. They are now clearer and in the main body of the paper (p. 6).

I like the study, I think it is important, and I am interested in what we will learn from it. I recommend that this research be accepted as a Registered Report. My only suggestion involves whether any of the sleep questionnaires collect information about caffeine consumption, and if not, whether that would be a worthwhile addition.

This is a good idea. It is standard practice in sleep studies to ask participants to refrain from consuming caffeine between the study session and the test session. We intend to follow this practice. However, we are aware that some participants forget to follow this instruction. We will therefore follow the reviewer's suggestion, and ask participants, after the test phase, about their caffeine consumption. This has been added to the paper (p. 11). We will run an exploratory analysis in our Stage 2 submission to see if caffeine intake modulates any of the results we find.

Reviewer: 2

Comments to the Author(s)

This is a very interesting study tackling a novel and important topic. Although the question whether eyewitness identification is affected by sleep is a very relevant practical issue, this question has not been formally tested yet. Findings on this question would also inform our general understanding of sleep and memory processes. The authors propose to examine two independent measures of eyewitness identification, i.e. discriminability and reliability. For discriminability they propose a specific hypothesis, i.e. they expect discriminability to be higher after sleep compared to a wake interval. This hypothesis is reasonable and based on previous evidence from sleep and recognition studies. However, for reliability the authors do not have a specific hypothesis based on lacking prior evidence. I would like to encourage the authors to state a hypothesis anyway, based on theoretical/conceptual considerations.

In a comprehensive review of the literature, Wixted and Wells (2017) conducted CAC analysis on a host of variables, including those that were once believed to affect reliability (e.g. short exposure durations, long retention intervals, the cross-race effect, etc.). In every case, reliability is not affected by these variables and instead high confidence responses were very high in accuracy and low confidence responses were much less so. This was the case whether exposure duration was long or short, retention interval was long or short, whether the perpetrator was the same or different race than the witness, etc. Because few variables appear to have an appreciable effect on high-confidence accuracy, it seems reasonable to suppose that the same might be true of sleep. We have added this to the manuscript (p. 6).

The proposed methods are adequate considering that this is a first broad study on the topic. Therefore it seems fine to obtain only very basic self-report information about sleep quality. However, if the authors want to draw sleep-specific conclusions, it would be important to test this question in a well-controlled environment with more fine-grained sleep measures, i.e. in a sleep lab with polysomnography. However, in a first step, as suggested in the present study, this will not be necessary, but the authors should be aware of the limited informational value of the sleep data they can obtain with questionnaires and this issue should also be discussed in the final paper.

We agree that collecting self-report information about sleep quality etc. does not enable us to draw sleep specific conclusions. Yet, as this reviewer notes, these methods can be used to test our hypothesis. To draw more sleep specific conclusions this study would need to be conducted in a sleep lab using polysomnography. This is not feasible because to conduct ROC and CAC analysis in a forensically relevant experiment (where there is only one trial per participant), large sample sizes (4,000 in this case) are needed. We will acknowledge this conundrum in the discussion of our Stage 2 submission.

The proposed statistical analyses are appropriate and well-established. Power estimations are difficult to run due to lacking previous evidence, but in my view the envisaged sample size seems sufficiently strong to detect at least medium size effects. The overall description of the methods is sufficiently clear for the authors not to use other undisclosed analyses or methods. However, I believe that minor changes that come up in the process of study preparation or data collection or analysis would be fine and can even be necessary.

The authors propose adequate controls for potential confounds and influencing factors. The two circadian (AM and PM) control groups are well suited to control for general circadian effects. Other relevant factors will be assessed via questionnaires, e.g. day-time sleepiness (ESS) and current sleepiness (SSS). This is all good. However, there are a few details that are not quite clear to me with regard to the control variables:

- Do the authors have a strategy for the case that there are circadian differences? How would the data in the sleep and wake groups be corrected?

Based on an abundance of research on sleep and memory that has used similar circadian controls to ours (e.g., Barrett & Ekstrand, 1972; Ellenbogen, Hulbert, Stickgold, Dinges & Thompson-Schill, 2006; Sheth, Nguyen & Janvelyan, 2009), we do not expect there to be a circadian confound that might be driving the predicted sleep benefit. However, if our circadian control conditions did reveal such an effect, this would be a very interesting discovery in its own right, and we would discuss it and its implications accordingly. We are not aware of a reliable method of *correcting* the data to take into account circadian effects. The best way then to find out whether sleep has a benefit over and above circadian effects would be to carry out further research in the laboratory, such as nap studies which fully control for time of day effects. But such work would be beyond the scope of the current experiment.

- Moreover, I am not sure when exactly the current sleepiness (SSS) is going to be assessed. At some place in the procedures it sounds like the wake group would not fill out the

SSS again during the test session. I would suggest obtaining two measures in the SSS in all groups (before study and after test).

This is a good point. We will collect the SSS in all conditions before the study phase and after the test phase. This has been added to the paper (p. 10-11).

- Related to the former point, the analysis strategy for the current sleepiness analysis should be refined. For this analysis, are the authors going to use the values from the study phase or test phase?

We intend to follow standard practice in the sleep research literature. Therefore SSS values will be compared across the sleep and wake groups in both phases as a manipulation check so we can rule out the possibility that sleepiness at encoding and/or retrieval is driving the results (p. 14).

- Finally, does the SMHS sleep questionnaire have an overall score? If the authors intend to use the single sleep quality item only, this may not represent a valid measure of sleep quality. Other measures such as sleep duration, number of awakenings etc. should also be included in the sleep quality analysis.

We agree that the question from the SMSH that we intend to use will not give us information that we could obtain in the sleep laboratory where we could obtain objective information about sleep duration and number of awakenings and arousals. Under the conditions of an online experiment however, we believe that the SMSH item coupled with participants' self-reported sleep duration will give us as reliable information about sleep quality as is possible to obtain. The analysis looking at the relationship between sleep quality and eyewitness identifications will be exploratory, and has now been removed from this submission at the request of the editor. It will be added to the Stage 2 submission after data collection.

The authors may consider introducing more exclusion criteria, e.g. psychiatric diagnoses (depression, PTSD?) and other medication (e.g. psychoactive drugs that may affect learning and memory). More questionnaires could be included to test for the exclusion criteria specifically instead of simply asking the subjects if these criteria apply to them. For example, the PSQI could be used to test for sleep disorders instead of simply asking "Are you currently diagnosed with a sleep disorder?" (many sleep disorders are not diagnosed and some subjects may not even know that they suffer from sleep disorders). Also, excluding subjects with less than 6 hours of sleep instead of 5 hours would be more reasonable.

In response to these suggestions we have revised the exclusionary criteria. Individuals who indicate that they are currently diagnosed with any psychiatric disorder, or who are currently taking medication that may impact their sleep will not be able to participate.

Following the reviewer's advice, participants in the sleep condition who have had less than 6 hours sleep between the study and test phase will be excluded from the analysis and replaced (this has been added to the manuscript). These points have been added to the paper (Sec.3).

Whilst the PSQI provides an indication of whether a participant has a sleep disorder, we hope to confirm this using the ESS. High scores on the ESS may be indicative of an underlying

sleep difficulties. The reason why we favour the ESS in this case is that the PSQI is too long and detailed to be filled in an online experiment.

Reviewer: 3

Comments to the Author(s)

SUMMARY

This proposed study is concerned with investigating the role of sleep in eyewitness identifications. Using measures of discriminability and reliability in a forensic identification setting, online participants will act as eyewitnesses through watching a video and attempt to identify the suspect in a subsequent photographic line-up. The interval between watching the video and line-up will contain either sleep or wake, with two further circadian control groups also present. A number of other sleep-related measures will be taken to investigate factors related to time of day, chronotype, sleep quality and sleepiness.

The proposed study is novel insofar as no eyewitness identification studies which look at performance as a function of sleep have been previously conducted, and it has clear real-world relevance. As such, it is a welcome study.

The design seems to be fundamentally sound, and most the proposed analyses are suitable for the intended purpose. The proposed sample size is large and the stimuli are appropriate. This promises to be an interesting and informative study, and I see no fundamental reason why it should not go ahead.

I do, however, have a number of suggestions for potential improvements, and in particular requests for clarification of two key areas of the Introduction (including in relation to the key measures to be used in the study), addressing issues surrounding online participation, and giving careful thought to other aspects of the design including the False ID rate calculation and the randomisation of the questionnaires. These and other minor points are detailed below:-

SPECIFIC POINTS (page, line/section)

1. 1, 40: “Not identify the innocent suspect” – I think you mean innocent fillers; the fillers are not suspects.

We do mean innocent suspect. Filler IDs are not as worrisome as innocent suspect IDs because fillers are known innocents whereas the police’s suspect is just that: their suspect. The eyewitness’s identification of that suspect can place more suspicion on that person or remove suspicion from him/her. We attached an illustrative figure here (and more information can be found in papers we cited in the manuscript, e.g. Wixted & Mickes, 2014, *Psychological Review*). Because fillers and the innocent suspects

are drawn from the same distribution in a fair lineup (the green distribution in the figure), we can estimate false ID rates by dividing the filler IDs from target-absent lineups by lineup size.

2. 1, 56-57: "Sleep's role in shaping reliability is completely unknown and our work will be the first to explore this variable". As a broad statement (that does not mention identification specifically), that is not sustainable, even in the context of eyewitness testimony. See for, example Blagrove and Akehurst (2000) ("Effects of sleep loss on confidence-accuracy relationships for reasoning and eyewitness memory"), which is unquestionably about eyewitness reliability and the role of sleep, and which, incidentally, is a curious omission from the current paper which should be rectified, given its obvious relevance. It is important to give due respect to prior research and not overstate novelty.

The Blagrove and Akehurst (2000) study measured the effects of sleep deprivation on confidence and accuracy on reasoning tasks, ravens matrices, and an acquiescence task, so this is much less related to our study than the title of their article would otherwise suggest. Furthermore they reported correlations, not calibration-type analyses, which further preclude us from making predictions about reliability based on their findings. We added the citation to our paper where we note that some work has been conducted on sleep deprivation and eyewitness memory (p. 5).

3. 2, 51: These appears to be some confusion over terminology and possible outcomes. To be clear, there are two scenarios – a suspect is present and identified (hit) or a suspect is present and not identified (miss). The authors seem to believe the innocence or guilt of the suspect creates two further scenarios where an "innocent suspect" is present in the line-up. That raises two questions: (1) How would the eyewitness know the suspect is innocent or guilty? (2) What is the difference between an "innocent suspect" and a filler? Objectively, either the suspect was present and witnessed at the crime scene, or he wasn't. If he was, we're into the original two scenarios. If he wasn't, then how he is different to all the other fillers who were also not present at the crime scene? If he was present at the crime scene but did not commit the crime, how is that a different eyewitness scenario in terms of memory, to him being present at the crime scene and committing the crime? In both cases the identification requirement is the same. I wonder if in fact the intention for the two further scenarios is just a line-up full of fillers? If this is the case, then say it's a line-up full of fillers and don't suggest there is an "innocent suspect" amongst them, which is paradoxical. In that case, also explain the forensic relevance of this, given that the police do not routinely run identity parades with no known suspects present. In short, I don't think these four outcomes have been properly thought through, or at least are not properly explained here.

A participant can make 1 of 3 possible responses: identify the suspect, identify a filler, or make no identification. Breaking that down further, if a participant is assigned to a target-present lineup, there are 3 possible responses: identify the guilty suspect, identify a filler, or make no identification. If a participant is assigned to a target-absent lineup, there are 3 possible responses: identify the innocent suspect, identify a filler, or make no identification.

From this, there are 3 possible outcomes from a decision made on a target-present lineup: correct ID (a hit), a filler ID, and a miss. And there are 3 possible outcomes from a decision made on a target-absent lineup: false ID, a filler ID, and a correct rejection. For a fair lineup

(where the suspect does not stand out amongst the fillers), the innocent suspect is effectively another filler (see the Figure above). This is why we can estimate the false ID rate the way we do (false ID rate = (filler IDs from target absent lineups/lineup size) / number of target absent lineups).

The reviewer asks, “*How would the eyewitness know the suspect is innocent or guilty?*” The eyewitness would have a memory trace for the guilty suspect because he was the perpetrator in the crime. But it is possible to false alarm (like claiming a lure word was on a list in a list learning recognition memory experiment).

The reviewer also asks, “*What is the difference between an ‘innocent suspect’ and a filler?*” Though an innocent suspect is effectively a filler in the lab (for a fair lineup), in the police station, this is not the case. The police have a suspect – who may be innocent or guilty. Fillers, on the other hand, are known to be innocent (which is why there is no risk that they would be wrongfully convicted if identified) and are there to populate the lineup. Of course, identifying the police suspect who happens to be innocent does imperil that person.

4. 3, throughout: The discussion about discriminability and reliability is similarly confusing and unconvincing. Discriminability is ability the “distinguish innocent from guilty suspects”. Operationally, that means measuring the proportion of correct (vs incorrect) identifications, which is of course equivalent to the probability that the identified suspect is the offender. Reliability, on the other, we are told is “the probability that the identified suspect is the offender”. In the other words, from the descriptions give, these are exactly the same measure. The examples given to illustrate that they are not the same measure don’t help because they both use the same sleight-of-hand, comparing discriminability on all cases with reliability on only high-confidence cases and declaring that they are different. Of course they are different – you have used a different dataset for each. If you used the same data, they wouldn’t be different, hence you have not demonstrated that these are different measures. If the measures are distinguished by the fact that one is only ever used in high-confidence cases while the other is always used in all cases, then the definitions should be given in those terms. As it is, I genuinely have no idea what, if anything, is actually the difference from the definitions given in the text. They seem to be the same to me. Given that they are referred to separately throughout the study, this should certainly be rewritten to address the above concerns and clarify what the actual difference is.

The reviewer writes, “*Discriminability is ability the ‘distinguish innocent from guilty suspects’. Operationally, that means measuring the proportion of correct (vs incorrect) identifications, which is of course equivalent to the probability that the identified suspect is the offender.*” This is incorrect. Discriminability and reliability are not the same measure. Although the distinction is subtle, they are mathematically very different.

We focus on high confidence responses when discussing reliability because witnesses who make IDs with high confidence are the witnesses who may end up in the court of law. We added a footnote to this effect (p. 4).

The reviewer writes, “*Of course they are different – you have used a different dataset for each*”, but this is untrue.

The distinction between reliability and discriminability can be easily appreciated by looking at the respective plots. This will be clear when we report our data graphically at Stage 2. For purposes of this review, we include a figure here. In this figure are hypothetical data from two conditions reported in Mickes (2016; Figure 4). You can see that, despite the same ROC curves where discriminability is lower in the verbal condition (as shown by lower ROC curves in A and B), positive predictive value (the measure of reliability, or CAC) can differ where it can be lower (B) or higher (D) in the verbal condition.

5. 6, participants: A good number of participants are specified, and I understand the lack of existing effect sizes to guide this, but you can actually turn this around and indicate the effect size that can be achieved with given alpha and beta levels, e.g. at the usual 0.05 alpha

rate and 95% power you could detect an effect size of $d=0.16$ in a direct comparison of sleep vs wake groups.

This is a great suggestion, which we have followed and included in the paper (p. 7).

6. 6, 34-48: Online participation is a sensible choice for such a large sample size, but it brings with it a host of issues around reliability (and mturk has recently . You should outline what quality control measures are going to be used to address these. For example, you specify 18-40 inclusion; how will you know that people really are 18-40? What have previous online studies found in regard to reliability of age reports? Or, for example, you want people in AM and PM conditions; what checks will be undertaken on digital time stamps to ensure that they really were in those conditions? Similarly, what about issues in regard to the different demographics of participants on mturk compared to the wider population (highlighted in a number of studies, e.g. Casler et al, 2013)? These are all things to consider and address at the design stage; they shouldn't be just disregarded.

Because forensically relevant experiments use only one trial per participant, and ROC and CAC analysis requires hundreds to thousands of participants to detect differences amongst groups or conditions, we (and others) have been collecting data online (e.g., Carlson & Carlson, 2014; Carlson et al., 2016; Colloff, Wade, & Strange, 2016; Gronlund et al., 2012; Mickes, 2015; Mickes, Flowe & Wixted, 2012; Mickes et al., in submission; Seale-Carlisle & Mickes, 2016; Wetmore et al., 2015; Wilson, Seale-Carlisle, & Mickes, in submission). In these studies, a validation question is included, which is typically "What crime was committed in the video?" Answering that correctly helps to ensure that attention was paid during the encoding phase. If answered incorrectly, the data will be excluded from analysis. In our experience, less than 2% answer incorrectly. These points have been added to the paper (p. 11-12).

Age:

Participants will not know that they have to be between 18-40 years to participate, so they will not be motivated to lie about their age. Furthermore, mTurk requires that workers must be at least 18 years of age. With such a large sample size, even if a few participants do lie about their age, they would do so in each condition, so random assignment will help mitigate this unlikely concern.

Digital Time Stamps:

The links provided to mTurk workers will only be active at certain times during the day, and the digital time stamp in the output files will confirm this.

Demographics:

Recruiting from the mTurk population actually broadens our sample in comparison to the student undergraduate population (which is often used in psychological research). Any study, for that matter, is effectively faced with concerns about generalizability.

7. 6, 60: I would suggest excluding shift workers as well, who will have a sleep pattern notably different to other participants, and possibly individuals with neurological disorders which could affect memory performance.

This is a good point. We revised our exclusionary criteria accordingly.

8. 7, 53: The distractor task had not been mentioned until this section, so either mention it in the Introduction, or explain its purpose here.

It is standard to include a distractor task to prevent rehearsal, and we added this to the paper (p. 9).

9. 9, 18: It's entirely up to the authors, but I would suggest thinking about whether you really want to randomise the order the questionnaires. Insofar as there will be order effects, you are doing comparisons within dependent measures across groups, i.e. you have not proposed multivariate analyses where randomisation might be necessary. Instead, randomisation here will have the effect only of increasing noise. For example, if sleepiness is increased during the session, putting the SSS in different locations during the test session will increase the variability of those responses, but this variability will not reflect actual differences in sleepiness at the point of the identification test (which is the implied use of the SSS), but rather only the different location at which the SSS was measured. If there is no difference across the session, then randomisation would make no difference. Either way, there is no clear benefit, and a potential downside, so if it were me, I would keep the procedure, including test order, consistent across all participants.

This is a good point. We will present the questionnaires in fixed order, and this change is in the paper (p. 11).

10. 10, 27: If the estimated False ID rate procedure is standard, or at least had good precedent, then I would cite previous papers that have used it here. I would also point out that the measure has some undesirable properties. In particular, by dividing by the number of lineup members, it effectively weights a correct identification at six times an incorrect identification, which makes combining the Correct ID and False IS measures problematic. I understand the reason for doing this – it makes chance level the same for both measures - but the more you deviate from chance, the more unbalanced the measures become (e.g. 50% misses and 50% false alarms don't give anything like equivalent values). This will be an issue when plotting the ROC pairs and conducting that analysis. In some sense, this goes to the question of behavioural equivalence – is a correct identification equivalent to an incorrect identification, or six incorrect identifications, and the answer to that is as much related to usage as to statistics. It's something to give careful thought to at least, even if in the end the existing measure is still preferred.

Using the estimated false ID rate is standard and does not differ when compared to other means to compute false ID rates (i.e. designating an innocent suspect in advance, randomly selecting an innocent suspect after data collection, using the filler chosen most often by participants; e.g. Seale-Carlisle & Mickes, 2016, *Royal Society Open Science*). We added references (p. 12).

We are unsure about what the reviewer's concerns are about weighting and why he considers chance performance 50% here, so we do not know how to respond to this.

Reviewer: 4

Comments to the Author(s)

The research question is highly important and will inform advice re treatment of victims and witnesses in terms of whether they are allowed to sleep before questioning.

The hypotheses are very plausible in terms of the considerable literature on effects of sleep on memory, and the results will have implications for that literature given that forensic-themed materials have been used only rarely in sleep and memory research.

The methodology and analyses are appropriate and are described in sufficient detail so as to allow exact replication.

The control condition of morning learning and evening testing is appropriate and a standard method, the methods will be able to test the proposed hypotheses.

Appendix B

Royal Holloway
University of London
Egham, Surrey
TW20 0EX

David Philip Morgan
PhD Candidate

Department of Psychology
David.morgan.2012@live.rhul.ac.uk
www.royalholloway.ac.uk

Professor Chris Chambers

27 August 2019

re: Response to Decision Letter

Dear Professor Chambers,

We are delighted to have completed data collection, analyses, and the write-up for the manuscript that we submitted two years ago entitled, "*The impact of sleep on eyewitness identifications.*"

To avoid being repetitive, please see the cover letter for details.

Sincerely,

David P. Morgan
Jakke Tamminen
Travis M Seale-Carlisle
Laura Mickes

Appendix C

Dear Professor Chambers,

We are delighted by the positive reviews and have addressed the concerns and made the relevant changes in the manuscript. Here is a brief list of the changes:

- Tense changed to past tense in the Stage 1 sections.
- A reviewer requested that we added statistical comparisons on the demographic information. These were added in a footnote and we made it clear that a reviewer requested them. It struck us as odd to include them as an exploratory analysis (which they are), but if you prefer that we move them to the exploratory analysis section, we will.
- Other changes were minor and indicated for ease of reading in blue colour font (except we did not change the colour of the tense changes given that there were so many).
- Added a footnote clarifying that participants were randomly assigned to AM or wake and PM or sleep conditions.
- Two of the authors changed affiliated institutions and this information has been updated.

We hope that these changes will make our paper ready for publication.

Sincerely,
David Morgan,
Jakke Tamminen,
Travis Seale-Carlisle,
Laura Mickes

11-Oct-2019

Dear Mr Morgan:

On behalf of the Editor, I am pleased to inform you that your Stage 2 Registered Report RSOS-170501.R1 entitled "The Impact of Sleep on Eyewitness Identifications" has been deemed suitable for publication in Royal Society Open Science subject to minor revision in accordance with the referee suggestions. Please find the referees' comments at the end of this email.

The reviewers and Subject Editor have recommended publication, but also suggest some minor revisions to your manuscript. Therefore, I invite you to respond to the comments and revise your manuscript.

Please also ensure that all the below editorial sections are included where appropriate -- if any section is not applicable to your manuscript, please can we ask you to nevertheless include the heading, but explicitly state that the heading is inapplicable. An example of these sections is attached with this email.

- Ethics statement

- Data accessibility

If you wish to submit your supporting data or code to Dryad (<http://datadryad.org/>), or modify your current submission to dryad, please use the following link:
<http://datadryad.org/submit?journalID=RSOS&manu=>(Document not available)

- Competing interests

- Authors' contributions

- Acknowledgements

- Funding statement

Because the schedule for publication is very tight, it is a condition of publication that you submit the revised version of your manuscript within 7 days (i.e. by the 19-Oct-2019). If you do not think you will be able to meet this date please let me know immediately.

Please note that Royal Society Open Science will introduce article processing charges for all new submissions received from 1 January 2018. Registered Reports submitted and accepted after this date will ONLY be subject to a charge if they subsequently progress to and are accepted as Stage 2 Registered Reports. If your manuscript is submitted and accepted for

publication after 1 January 2018 (i.e. as a full Stage 2 Registered Report), you will be asked to pay the article processing charge, unless you request a waiver and this is approved by Royal Society Publishing. You can find out more about the charges at <http://rsos.royalsocietypublishing.org/page/charges>. Should you have any queries, please contact openscience@royalsociety.org.

Kind regards,
Anita Kristiansen
Royal Society Open Science
openscience@royalsociety.org

on behalf of Chris Chambers (Registered Reports Editor, Royal Society Open Science)
openscience@royalsociety.org

Associate Editor Comments to Author (Professor Chris Chambers):

Associate Editor: 1

Comments to the Author:

Two of the four expert reviewers who assessed the Stage 1 manuscript have now appraised the completed Stage 2 submission. The reviews are overall very positive, with only minor revisions recommended. Reviewer 1 requests some changes to resolve issues of clarity. Reviewer 2 recommends a shift from future to past tense in the Introduction and Method sections. This is an acceptable alteration and one that I would also recommend in the interests of readability (please ensure that all changes are tracked in the revised manuscript). The reviewer also suggests an additional analysis (Reviewer 2, point 6). Additional exploratory analyses are not required for Stage 2 acceptance, except where necessary to support the conclusions. In this case, these analyses do not seem necessary for the current conclusions to be appropriate, therefore it is up to the authors to decide if the analyses would be appropriate and informative. If so, the authors are welcome to conduct these extra analyses and report them in the Exploratory section of the Results.

Reviewer: 1

Comments to the Author(s)

The authors examined whether sleep can benefit eyewitness discriminability and reliability using a forensically-relevant experiment. Prior research on sleep (positive impact on memory for faces, locations) suggests it should benefit discriminability, but no prior work has examined reliability. If the authors had found positive impacts on discriminability, or differential effects on reliability, the findings would have been of practical significance. However, even the null effects here are informative, suggesting that the timing of a lineup test with regard to sleep does not appear to affect performance.

The authors made use of an AM-PM-PM-AM sleep design plus two circadian control conditions. The data were analyzed appropriately (and presented clearly) using ROC and CAC analyses. The conclusions are clear, if not terribly exciting. The fact that there was a significant difference detected in Figure 3 (control/immediate test vs. experimental/delayed test) shows that the authors had sufficient power to detect reasonably-size differences. The one concern I have is that overall performance is quite good. I am not concerned about ceiling effects, but if the authors had to do it over again (and I am NOT suggesting that), I would have preferred a little lower level of performance.

On p. 15 (and a couple of places thereafter), the authors referred to a false ID cutoff. I believe they meant filler ID cutoff.

>> We thank the reviewer for noticing this. We have swapped the words “false ID” with “TA filler ID” where needed.

On p. 23, the authors wrote: Despite these similarities, there were differences: we 1) did not use the same stimuli or procedures, 2) DID NOT conducted separate experiment (CAPS added). As written, makes it sound like the authors conducted separate experiments for target-present and target-absent, but it was Stepan et al. that did this.

>> We thank the reviewer for noticing this too. We have added the critical words “did not” to the statement.

1. Whether the data are able to test the authors’ proposed hypotheses by passing the approved outcome-neutral criteria (such as absence of floor and ceiling effects or success of positive controls)

No floor or ceiling effects. Control groups appropriate and resulting data informative.

2. Whether the Introduction, rationale and stated hypotheses are the same as the approved Stage 1 submission

Yes

3. Whether the authors adhered precisely to the registered experimental procedures

Yes

4. Where applicable, whether any unregistered exploratory statistical analyses are justified, methodologically sound, and informative

I appreciated the exploratory statistical analyses that were conducted and found them all to be informative.

5. Whether the authors’ conclusions are justified given the data

Yes

Comments to Author:

Reviewer: 2

Comments to the Author(s)

I have reviewed this manuscript before as a registered report prior to data collection. As already expressed in my first assessment of the manuscript, I feel that the study addresses a novel and important topic; the methods are sound and the proposed data analyses are appropriate. The authors have also included my previous suggestions on minor refinements

of their protocol and analyses. This is all very good.

I would like to congratulate the authors on this comprehensive study, including such a large sample size. As far as I can see, the authors adequately adhered to their proposed experimental procedures as well as the planned statistical analyses. Beyond that, they have included appropriate exploratory analyses, which are clearly labelled as such.

The findings are very interesting in showing that, contrary to the hypothesis, sleep did not affect eyewitness identification (neither discriminability nor reliability). Given that the sample size was large enough to detect small effects, this is a very convincing demonstration that sleep may not influence the identification of perpetrators in lineups.

These findings are very informative and practically relevant. The authors appropriately discuss possible explanations for this null finding and discuss potential implications.

There are only a few minor issues in the writing of the manuscript and the description of the methods that should be addressed in a revision:

1) I am not entirely clear on the style of a registered report paper. As is written now, the manuscript basically includes two separate parts, the first part including the introduction, methods and planned analyses as was submitted in the Stage 1 review. The second part includes the results and discussion sections, which were simply added to the first part. The two parts are clearly distinguishable, as the first part is written in future tense and the second in past tense. In the second part, some statements refer to other statements from the first part that have changed in the meantime (e.g. a new study on the same topic that has been published in the meantime). In my feeling, it would be more appropriate to re-write the entire paper as one single report, written in past tense. This would make the manuscript much more accessible for the reader. The abstract should also mention the results and a short conclusion statement.

>> We agree with the reviewer that reading the manuscript in its entirety is awkward because of the tense change and we appreciate that the editor allowed us to make it consistent. We have done so.

2) The lineup procedure is not entirely clear to me. Was each subject presented with only one lineup? Did half of the subjects receive a target-present lineup and the other half a target-absent lineup? I was confused by the statement that “we... conducted separate experiments for target-absent and target-present lineups” (page 23). Please clarify and describe more clearly in the methods section.

>> We fixed the error that we made on page 23 (also noticed by Reviewer 1). We definitely did not conduct target-absent and target-present lineups in separate experiments.

>> We wrote in the Introduction of the Stage 1 submission, “there is only one trial per participant in a forensically relevant design (to mimic the experience of a real eyewitness)”.

>> The number of participants assigned to a target-absent or target-present lineups is presented in Table 2 for sleep and wake conditions and in Table 3 for AM and PM control conditions in the “Total” rows.

3) Please explain in more detail what the most conservative overall false ID rate in the ROC analyses refers to and how it was determined.

>> We added more explanation about the most conservative overall false ID rate being the rightmost point from the condition that yielded more conservative responding overall.

4) Table 1 should include statistical comparisons between the single groups to show that all groups were comparable in demographic features.

>> We did the analysis suggested by the reviewer and we added this as a footnote in the paper, "A reviewer of the Stage 2 manuscript recommended that we conduct analysis on differences in demographics between conditions. There were no significant differences in ethnicity [$\chi^2(12) = 1.55, p = .100$] (chi-square limited to ethnicities with cells > 5), gender [$\chi^2(3) = .61, p = .611$], or education [$\chi^2(12) = 1.61, p = .080$]. There were significant differences in age between conditions [$F(3,3996) = 9.18, p < .001$], but that is because the sample size is so large. The effect size is trivially small (.007). An age difference of .85 (30.40 - 29.55 average years of the wake and sleep groups, respectively) is unlikely to account for results."

5) The terms "filler IDs" and "no-IDs" are mentioned in the results section and in the Tables and Figures, yet these terms are not introduced in the methods section. Are filler IDs identical to false IDs? Please clarify and include descriptions in the methods section.

>> In one sense they are the same because having not been presented during the study phase, they are all "new" to the participants. However, filler IDs are not practically the same because people who serve as fillers are known to be innocent by the police. Thus, for applied purposes, it's false IDs (incorrect suspect IDs) and correct IDs (correct suspect IDs) that are of most interest. We hesitate to add this to the Method section as this was covered in the introduction and was approved during Stage 1.

6) With regard to the exploratory analyses, it would also be interesting to test for the effects of habitual sleepiness (ESS), chronotype and total sleep time in the experimental night. The authors may consider adding respective analyses.

>> We are not sure what question this analysis would be addressing, so we will not include it in the manuscript. For researchers interested in conducting this analysis, they can do so with the data that are available.

7) In my view, the additional analyses in Figure 3 are particularly important since it is not a trivial question that discriminability declines across a 12 hour delay. This may be discussed in more detail as it is a practically relevant issue. It is also an interesting finding that reliability does not seem to deteriorate across time (at least across 12 hours). Please report statistics for the latter finding (page 23). The term "similarly well-calibrated" sounds unusual to me and could be rephrased.

>> We are not sure that we understand what the reviewer means by "...not a trivial question that discriminability declines across a 12 hour delay..." It has been well-known for a long time that forgetting follows a decaying mathematical function, and that the decrease starts immediately after learning (e.g., Ebbinghaus, 1885). We would be puzzled if there was not a difference in discriminability between conditions that had a 5-minute vs. 12-hour retention

interval. Because we think that we must be misunderstanding this point, we will have to leave this statement.

>> CAC analysis is a graphical analysis. We used a bootstrapping procedure to calculate standard errors and use non-overlapping error bars to consider differences. This has precedence (all of the CAC plots have been presented in this manner; e.g., Mickes, 2015; Wixted & Wells, 2017). We specified this in our Stage 1 submission. In this case, the error bars are not overlapping, so they are similarly well-calibrated. To make this clearer, we added this statement to the paper, "...participants were similarly well-calibrated. That is, in the experimental and control conditions, identifications made with high confidence are higher in accuracy than identifications made with medium confidence which are higher in accuracy than identifications made with low confidence (and the standard error bars are overlapping)."

8) Typos: "some aspects eyewitness memory" (page 5), "higher than accuracy then low confidence identifications" (page 23), in reference #63 two references seem to be mixed up, in Table 3 it should probably read "for AM and PM controls", in Figure 3B it should probably read "for experimental and control groups"

>> We thank the reviewer for catching these typos and have fixed them.